# Language Models Linearly Represent Sentiment

## Abstract

Sentiment is a pervasive feature in natural language text, yet it is an open question how sentiment is represented within Large Language Models (LLMs). In this study, we reveal that across a range of models, sentiment is represented linearly: a single direction in activation space mostly captures the feature across a range of tasks with one extreme for positive and the other for negative. Through causal interventions, we isolate this direction and show it is causally relevant in both toy tasks and real world datasets such as Stanford Sentiment Treebank.

We further uncover the mechanisms that involve this direction, highlighting the roles of a small subset of attention heads and neurons. Finally, we discover a phenomenon which we term the summarization motif: sentiment is not solely represented on emotionally charged words, but is additionally summarised at intermediate positions without inherent sentiment, such as punctuation and names. We show that in Stanford Sentiment Treebank zero-shot classification, ablating the sentiment direction across all tokens results in a drop in accuracy from 100% to 62%, while ablating the summarized sentiment direction at comma positions alone produces close to half this result (reducing accuracy to 82%).

## 1 Introduction

Large language models (LLMs) have displayed increasingly impressive capabilities (Brown et al., 2020; Radford et al., 2019; Bubeck et al., 2023), but their internal workings remain poorly understood. Nevertheless, recent evidence (Li et al., 2023) has suggested that LLMs are capable of forming models of the world, i.e., inferring hidden variables of the data generation process rather than simply modeling surface word co-occurrence statistics. There is significant interest (Christiano et al. (2021), Burns et al. (2022)) in deciphering the latent structure of such representations.

In this work, we investigate how LLMs represent sentiment, a variable in the data generation process that is relevant and interesting across a wide variety of language tasks(Cui et al., 2023). Approaching our investigations through the frame of causal mediation analysis (Vig et al., 2020; Pearl, 2022), we show that these sentiment features are represented linearly by the models, are causally significant, and are utilized by human-interpretable circuits (Olah et al., 2020; Elhage et al., 2021a).

We find the existence of a single direction scientifically interesting as further evidence for the linear representation hypothesis (Mikolov et al., 2013; Elhage et al., 2022)–the word2vec hypothesis, that models tend to extract properties of the input and internally represent them as directions in activation space. Understanding the structure of internal representations is crucial to begin to decode them, and linear representations are particularly amenable to detailed reverse-engineering (Nanda et al., 2023b).

We show evidence of a phenomenon we have labeled the "summarization motif", where rather than sentiment being directly moved from valenced tokens to the final token, it is first aggregated on intermediate summarization tokens without inherent valence such as commas, periods and particular nouns. This can be seen as a naturally emerging analogue to the explicit classification token in BERT-like models (Devlin et al., 2018), and in that context the phenomenon was observed by Clark et al. (2019). We show that the sentiment stored on summarization tokens is causally relevant for the final prediction. We find this an intriguing example of an "information bottleneck", where the data generation process is funnelled through a small subset of tokens used as information stores.

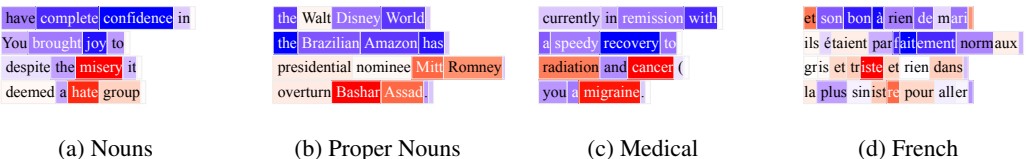

(a) Nouns          (b) Proper Nouns          (c) Medical          (d) French

Figure 1: Visualizing the "sentiment activation" (projection of the residual stream onto the sentiment axis) where blue is positive and red is negative. Examples (1a-1c) show the $k$-means sentiment direction for the first layer of GPT2-small on samples from OpenWebText. Example 1d shows the $k$-means sentiment direction for the 7th layer of pythia-1.4b on the opening of Harry Potter in French.

Understanding the existence and location of information bottlenecks is a key first step to deciphering world models. This finding additionally suggests the models' ability to create summaries at various levels of abstraction, in this case a sentence or clause rather than a token.

Our contributions are as follows. In Section 3, we demonstrate methods for finding a **linear representation of sentiment** using a toy dataset and show that this direction correlates with sentiment information in the wild and matters causally in a crowdsourced dataset. In Section 4, we show through activation patching (Vig et al., 2020) and ablations (techniques defined in Section 2.3) that the learned sentiment direction captures **summarization behavior** that is causally important to circuits performing sentiment tasks.

## 2 METHODS

### 2.1 DATASETS AND MODELS

**ToyMovieReview**    is a templatic dataset of continuation prompts we generated with the form

> I thought this movie was ADJECTIVE, I VERBed it. Conclusion: This movie is

where ADJECTIVE  and VERB  are either two positive words (e.g., incredible and enjoyed) or two negative words (e.g., horrible and hated) that are sampled from a fixed pool of 85 adjectives (split 55/30 for train/test) and 8 verbs. The expected completion for a positive review is one of a set of positive descriptors we selected from among the most common completions (e.g.  great) and the expected completion for a negative review is a similar set of negative descriptors (e.g., terrible).

**ToyMoodStory**    is a similar toy dataset which is multi-subject and character-driven with random names, e.g. Carl hates parties, and avoids them whenever possible. Jack loves parties, and joins them whenever possible. One day, they were invited to a grand gala. Jack feels very [excited/nervous]

**Stanford Sentiment Treebank (SST)**    Socher et al. (2013) consists of 10,662 one sentence movie reviews with human annotated sentiment labels for every constituent phrase from every review.

**Internet Movie Database (IMDB)**    Maas et al. (2011) consists of 25,000 movie reviews taken from the IMDB website with human-annotated sentiment labels for each review.

**OpenWebText**    (Gokaslan & Cohen, 2019) is the pretraining dataset for GPT-2 which we use as a source of random text for correlational evaluations.

**GPT-2 and Pythia**    (Radford et al., 2019; Biderman et al., 2023) are families of decoder-only transformer models with sizes varying from 85M to 2.8b parameters. We use GPT2-small for movie review continuation, pythia-1.4b for classification and pythia-2.8b for multi-subject tasks.

### 2.2 FINDING DIRECTIONS

We use five methods to find a sentiment direction in each layer of a language model using our ToyMovieReview dataset. In each of the following, let $\mathbb{P}$ be the set of positive inputs and $\mathbb{N}$ be the

set of negative inputs. For some input $x \in \mathbb{P} \cup \mathbb{N}$, let $\boldsymbol{a}_x^L$ and $\boldsymbol{v}_x^L$ be the vector in the residual stream at layer $L$ above the adjective and verb respectively. We reserve $\{\boldsymbol{v}_x^L\}$ as a hold-out set for testing. Let the correct next token for $\mathbb{P}$ be $p$ and for $\mathbb{N}$ be $n$.

**Mean Difference (MD)**    The direction is computed as $\frac{1}{|\mathbb{P}|} \sum_{p \in \mathbb{P}} \boldsymbol{a}_p^L - \frac{1}{|\mathbb{N}|} \sum_{n \in \mathbb{N}} \boldsymbol{a}_n^L$.

$k$-**means (KM)**    We fit 2-means to $\{\boldsymbol{a}_x^L : x \in \mathbb{P} \cup \mathbb{N}\}$, obtaining cluster centroids $\{\boldsymbol{c}_i : i \in [0, 1]\}$ and take the direction $\boldsymbol{c}_1 - \boldsymbol{c}_0$.

**Linear Probing**    The direction is the normed weights $\frac{\boldsymbol{w}}{||\boldsymbol{w}||}$ of a logistic regression (**LR**) classifier $\mathbb{P}_{\boldsymbol{w}}(a_x^L) = \frac{1}{1 + \exp(-\boldsymbol{w} \cdot \boldsymbol{a}_x^L)}$ trained to distinguish between $x \in \mathbb{P}$ and $x \in \mathbb{N}$.

**Distributed Alignment Search (DAS)**    The direction is a learned parameter $\theta$ where the training objective is the average logit difference

$$\sum_{x \in \mathbb{P}} [\text{logit}_\theta(x; p) - \text{logit}_\theta(x; n)] + \sum_{x \in \mathbb{N}} [\text{logit}_\theta(x; n) - \text{logit}_\theta(x; p)]$$

after activation patching (a technique outlined in Section 2.3) using direction $\theta$ .

**Principal Component Analysis (PCA)**    The direction is the first component of $\{\boldsymbol{a}_x^L : x \in \mathbb{P} \cup \mathbb{N}\}$.

## 2.3 CAUSAL INTERVENTIONS

**Activation patching**    In activation patching (Geiger et al., 2020; Vig et al., 2020), we create two symmetrical datasets, where each prompt $x_{\text{orig}}$ and its counterpart prompt $x_{\text{flipped}}$ are of the same length and format but where key words are changed in order to flip the sentiment; e.g., "This movie was great" could be paired with "This movie was terrible". We first conduct a forward pass using $x_{\text{orig}}$ and capture these activations for the entire model. We then conduct forward passes using $x_{\text{flipped}}$, iteratively patching in activations from the original forward pass for each model component. We can thus determine the relative importance of various parts of the model with respect to the task currently being performed.

Geiger et al. (2023b) introduce a variant of activation patching that we call "directional activation patching". The idea is that rather than modifying the standard basis directions of a component, we instead only modify the component along a single direction in the vector space, replacing it during a forward pass with the value from a different input.

Another variant is path patching (Wang et al., 2022) in which only the activations related to the residual stream paths between two sets of endpoints (senders and receivers) are patched, but the remainder of the network upstream of the receivers is frozen.

We use two evaluation metrics. The logit difference (difference in logits for correct and incorrect answers) metric introduced in Wang et al. (2022), as well as a "logit flip" metric (Geiger et al., 2022), which quantifies the proportion of cases where we induce an inversion in the predicted sentiment.

**Ablations**    We eliminate the contribution of a particular component to a model's output, usually by replacing the component's output with zeros (zero ablation) or the mean over some dataset (mean ablation), in order to demonstrate its magnitude of importance. We also perform directional ablation, in which a component's activations are ablated only along a specific (e.g. sentiment) direction.

## 3 FINDING AND EVALUATING A 'SENTIMENT DIRECTION'

The first question we investigate is whether there exists a direction in the residual stream in a transformer model that represents the sentiment of the input text, as a special case of the linear representation hypothesis (Mikolov et al., 2013). We show that the methods discussed above (2.2) all arrive at a similar sentiment direction. We can visualise the feature being represented by this direction by projecting the residual stream at a given token/layer onto it, using some text from the training distribution. We will call this the "**sentiment activation**".

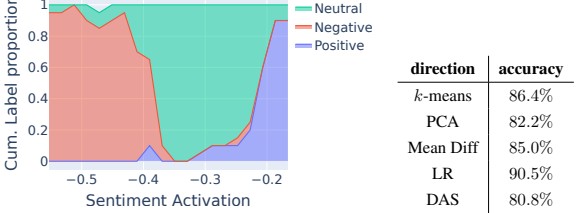

|        | DAS    | KM     | LR     | MD     | PCA    | Random |
|--------|--------|--------|--------|--------|--------|--------|
| DAS    | 100.0% | 82.4%  | 86.1%  | 86.1%  | 69.7%  | 0.2%   |
| KM     | 82.4%  | 100.0% | 95.1%  | 95.7%  | 80.0%  | 1.7%   |
| LR     | 86.1%  | 95.1%  | 100.0% | 99.9%  | 78.0%  | 0.6%   |
| MD     | 86.1%  | 95.7%  | 99.9%  | 100.0% | 79.5%  | 0.6%   |
| PCA    | 69.7%  | 80.0%  | 78.0%  | 79.5%  | 100.0% | 0.7%   |
| Random | 0.2%   | 1.7%   | 0.6%   | 0.6%   | 0.7%   | 100.0% |

| direction | accuracy |
|-----------|----------|
| $k$-means | 86.4%    |
| PCA       | 82.2%    |
| Mean Diff | 85.0%    |
| LR        | 90.5%    |
| DAS       | 80.8%    |

(a) Cosine similarity of directions learned by different methods in pythia-2.8b's first layer. Each sentiment direction was derived from *adjective* representations in the ToyMovieReview dataset (Section 2.1).

(b) Area plot of sentiment labels for OpenWebText samples by sentiment activation, i.e. the projection of the first residual stream layer of pythia-2.8b at that token onto the sentiment direction (left). Accuracy using sentiment activations to classify tokens as positive or negative (right). The threshold taken is the top/bottom 0.1% of activations over OpenWebText. Classification was performed by GPT-4.

|                     | simple_logit_diff | treebank_logit_diff | simple_logit_flip | treebank_logit_flip |
|---------------------|-------------------|---------------------|-------------------|---------------------|
| das                 | 109.8%            | 47.0%               | 100.0%            | 53.5%               |
| das2d               | 110.4%            | 42.8%               | 95.5%             | 49.0%               |
| das3d               | 110.2%            | 35.9%               | 95.5%             | 39.4%               |
| kmeans              | 67.2%             | 22.1%               | 72.7%             | 14.8%               |
| logistic_regression | 71.1%             | 30.8%               | 86.4%             | 16.8%               |
| mean_diff           | 73.9%             | 27.5%               | 81.8%             | 17.4%               |
| pca                 | 62.7%             | 17.8%               | 72.7%             | 12.3%               |
| random              | 0.4%              | 0.1%                | 0.0%              | 0.6%                |

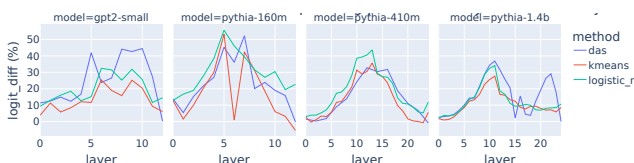

(c) Directional patching results for different methods in pythia-1.4b (2.8b not shown due to compute time). We report the best result found across layers. The columns show two evaluation datasets, ToyMovieReview and Treebank, and two evaluation metrics, mean logit difference and % of logit differences flipped.

(d) Patching results for directions trained on toy datasets and evaluated on the Stanford Sentiment Treebank test partition. We tend to find the best generalisation when training and evaluating at a layer near the middle of the model. We scaffold the prompt using the suffix Overall the movie was very and compute the logit difference between good and bad. The patching metric (y-axis) is then the % mean change in logit difference.

Figure 2: A correlational and causal analysis of sentiment directions.

## 3.1 COMPARING THE DIRECTIONS

We fit directions using the residual stream over the adjective token in the ToyMovieReview dataset (Section 2.1) and the methods outlined in Section 2.2, finding extremely high cosine similarity (Figure 2a). This suggests that these are all noisy approximations of the same singular direction. Indeed, we generally found that the following results were very similar regardless of exactly how we specified the sentiment direction.

## 3.2 CORRELATIONAL EVALUATION

**Visualizing The Sentiment Direction** Here we show a visualisation in the style of Neuroscope (Nanda, 2023a) where the sentiment activation (the projection of the residual stream onto the sentiment axis) is represented by color, with red being negative and blue being positive. It is important to note that the direction being examined here was trained on just 30 positive and 30 negative English adjectives in an unsupervised way (using $k$-means with $k = 2$). Notwithstanding, the extreme values along this direction appear readily interpretable in the wild in diverse text domains such as the opening paragraphs of Harry Potter in French (Figure 1).

**Quantifying classification accuracy** To rigorously validate this visual check, we binned the sentiment activations of OpenWebText tokens from the first residual stream layer of GPT2-small into 20 equal-width buckets and sampled 20 tokens from each. Then we asked GPT-4 to classify into Positive/Neutral/Negative. [1] In Figure 2b, we show an area plot of the classifications by activation bin. We contrast the results for different methods in Table 2b. In the area plot we can see that the left side area is dominated by the "Negative" label, whereas the right side area is dominated by the

---

[1] Specifically, we gave the GPT-4 API prompts of the following form: "Your job is to classify the sentiment of a given token (i.e. word or word fragment) into Positive/Somewhat positive/Neutral/Somewhat negative/Negative. Token: '{token}'. Context: '{context}'. Sentiment: " where the context length was 20 tokens centred around the sampled token.

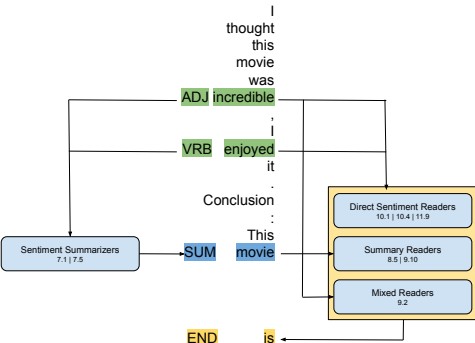

Figure 3: Primary components of GPT-2 sentiment circuit for the ToyMovieReview dataset. Here we can see both direct use of sentiment-laden words in predicting sentiment at END as well as an example of the summarization motif at the SUM position (the final 'movie' token). Heads 7.1 and 7.5 write to this position and this information is causally relevant to the contribution of the summary readers at END.

"Positive" label and the central area is dominated by the "Neutral" label. Hence the tails of the activations seem highly interpretable as representing a bipolar sentiment feature. The large space in the middle of the distribution simply occupied by neutral words (rather than a more continuous degradation of positive/negative) indicates superposition of features (Elhage et al., 2022).

## 3.3 CAUSAL EVALUATION

**Sentiment directions are causal representations.**   We evaluate the sentiment direction using directional patching (Section 2.3) in Figure 2c. These evaluations are performed on prompts with out-of-sample adjectives and the direction was not trained on *any* verbs. Unsupervised methods such as $k$-means are still able to shift the logit differences and DAS is able to completely flip the prediction.

**Directions Generalize Most at Intermediate Layers**   If the sentiment direction was simply a trivial feature of the token embedding, then one might expect that directional patching would be most effective in the first or final layer. However, we see in Figure 2d that in fact it is in intermediate layers of the model where we see the strongest out-of-distribution performance on SST. This suggests the speculative hypothesis that the model uses the residual stream to form abstract concepts in intermediate layers and this is where the latent knowledge of sentiment is most prominent.

**Validation on SST**   We validate our sentiment directions derived from toy datasets (Section 3.3) on SST. We collapsed the labels down to a binary "Positive"/"Negative", just used the unique phrases rather than any information about their source sentences, restricted to the 'test' partition and took a subset where pythia-1.4b can achieve 100% zero shot classification accuracy, removing 17% of examples. Then we paired up phrases of an equal number of tokens[2] to make up 460 clean/corrupted pairs. We used the scaffolding "Review Text: TEXT, Review Sentiment:" and evaluated the logit difference between "Positive" and "Negative" as our patching metric. Using the same DAS direction from Section 3 trained on just a few examples and flipping the corresponding sentiment activation between clean/corrupted in a single layer, we can flip the output 53.5% of the time (Figure 2c).

**Validation at the document level**   In order to verify the applicability of our findings to larger document-sized prompts, we performed directional ablation (as described in 2.3) on the IMDB dataset, most of which consists of multiple sentences. Each item of this dataset was appended with "Review Sentiment:" in order to prompt a classification completion, and we selected 1000 examples each from the positive and negative items that the model was capable of classifying correctly.

---

[2]We did this to maximise the chances of sentiment tokens occurring at similar positions

We used the sentiment directions found with DAS to ablate sentiment at every token at every layer (using Pythia-2.8b). As a result, sentiment classification accuracy dropped from 100% to 57%.

# 4 THE SUMMARIZATION MOTIF FOR SENTIMENT

## 4.1 CIRCUIT ANALYSES

In this sub-section, we present an overview of circuit[3] findings that give qualitative hints of the summarization motif, and restrict quantitative analysis of the summarization motif to 4.2. Through an iterative process of path patching (see Section 2.3) and analyzing attention patterns, we have identified key components of the circuit responsible for the ToyMovieReview task in GPT2-small (Figure 3) as well as the circuit for the ToyMoodStories task in Pythia-2.8b. Below, we provide a brief overview of the circuits we identified, reserving the full details for A.3.

**Initial observations of summarization in GPT-2 circuit for ToyMovieReview**    Mechanistically, this is a binary classification task, and a naive hypothesis is that attention heads attend directly from the final token (which we label 'END') to the valenced tokens (the adjective token, ADJ, and the verb token VRB) and map positive sentiment to positive outputs and vice versa. This does happen but it is not the only mechanism. Attention head output is causally important at intermediate token positions (in particular, the final 'movie' token, SUM), which are then read from when producing output at END. We consider this an instance of summarization, in which the model aggregates causally-important information relating to an entity at a particular token for later usage, rather than simply attending back to the original tokens that were the source of the information.

To summarize our findings, we find that the model implements a simple, interpretable procedure to perform the task (using a circuit made up of 9 attention heads). We used the path patching technique mentioned above to find and validate this circuit and the procedure it implements, as detailed in Appendix A.3:

1. Identify sentiment-laden words in the prompt, at ADJ and VRB.
2. Write out sentiment information to SUM (the final "movie" token).
3. Read from ADJ, VRB and SUM and write to END.[4]

The results of activation patching the residual stream can be seen in the Appendix, Fig. A.8. For a subset of the heads, the output of the attention is only important at the movie token, which we designate as the SUM position. We label these heads "sentiment summarizers." Specific attention heads ("direct effect heads") attend to and rely on information written to this token position as well as to ADJ and VRB.

To further validate this circuit and the involvement of the sentiment direction, we patched the entirety of the circuit at the ADJ and VRB positions along the sentiment direction only, achieving a 58.3% rate of logit flips and a logit difference drop of 54.8% (in terms of whether a positive or negative next token was predicted). Patching the circuit at those positions along all directions resulted in flipping 97% of logits and a logit difference drop of 75%, showing that the sentiment direction is responsible for the majority of the function of the circuit.

**The ToyMoodStory task in Pythia-2.8b**    We next examined the circuit that processes the Toy-MoodStory dataset (Section 2.1) in Pythia-2.8b, the smallest model that could perform this more complex task that requires more summarization. We reserve a detailed description for the Appendix (4.1), but note here that we observed increasing reliance on summarization, specifically:

- "Comma-reading heads": A set of attention heads **attended primarily to the comma** following the preference phrase for the queried subject (e.g. John hates parties,), and secondarily to other

---

[3]We use the term "circuit" as defined by Wang et al. (2022), in the sense of a computational subgraph that is responsible for a significant proportion of the behavior of a neural network on some predefined task.

[4]We note that our patching experiments indicate that there is no causal dependence on the output of other model components at the ADJ and VRB positions–only at the SUM position.

[5]That is, the attention pattern weighted by the norm of the value vector at each position as per Kobayashi et al. (2020). We favor this over the raw attention pattern as it filters for *significant* information being moved.

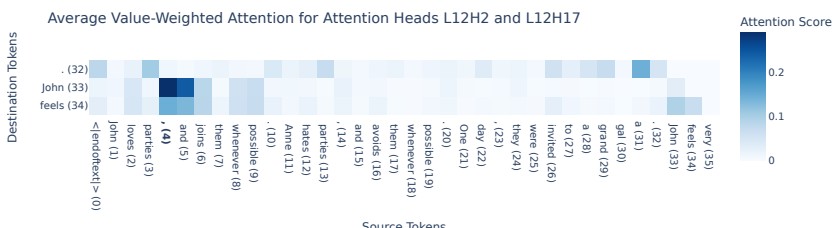

Figure 4: Value-weighted[5] averaged attention to commas and comma phrases in Pythia-2.8b from the top two attention heads writing to the repeated name and "feels" tokens–two key components of the summarization sub-circuit in the ToyMoodStories task. Note that they attend heavily to the relevant comma from both destination positions.

words in the phrase, as seen in Figure 4. We observed this phenomenon both with regular attention and value-weighted attention, and found via path patching that **these heads relied primarily on the comma token** for their function, as seen in Figure A.10.

- "Name-writing heads": Heads attending to preference phrases (e.g., the entirety of "John loves parties," including the final comma) tended to write to the repeated name token near the end of the sentence (John) as well as to the feels token–another type of summarization behavior.

- "Name-reading heads": Later heads attended to the repeated name and feels tokens, affecting the output logits at END.

## 4.2 EXPLORING AND VALIDATING SUMMARIZATION BEHAVIOR IN PUNCTUATION

Our circuit analyses reveal suggestive evidence that summarization behavior at intermediate tokens like commas, periods and certain nouns plays an important part in sentiment processing, despite these tokens having no inherent valence. We focus on summarization at commas and periods and explore this further in a series of ablation and patching experiments. We find that in many cases this summarization results in a partial information bottleneck, in which the summarization points become as important (or sometimes more important) than the phrases that precede them for sentiment tasks. Although we continue to focus on results from Pythia-2.8b, we include results from other models that substantiate our findings in Appendix A.4.2.

**Summarization information is comparably important as original semantic information**  In order to determine the extent of the information bottleneck presented by commas in sentiment processing, we tested the model's performance on ToyMoodStory (Section 2.1). We froze the model's attention patterns to ensure the model used the information from the patched commas in exactly the same way as it would have used the original information. Without this step, the model could simply avoid attending to the commas. We then performed activation patching on either the pre-comma phrases (e.g., patching "John hates parties," with "John loves parties,") while freezing the commas and periods so they retain their original, unflipped values; or on the two commas and two periods alone. The results show a similar drop in the logit difference for both cases, as indicated in Table 1a. Results for other models can be seen in A.4.2.

Table 1: Patching in ToyMoodStory

| Intervention | Change in logit difference |
|---|---|
| Patching full phrase values (incl. commas) | -75% |
| Patching pre-comma values (freezing commas & periods) | -38% |
| Patching comma and period values only | -37% |

(a) Change in logit difference from intervention on attention head value vectors in Pythia 2.8b

| Count of irrelevant tokens after preference phrase | Ratio of LD change for periods vs. phrases |
|---|---|
| 0 tokens | 0.29 |
| 10 tokens | 0.63 |
| 18 tokens | 0.92 |
| 22 tokens | 1.15 |

(b) Ratio between logit difference change for periods vs. pre-period phrases after patching values

**Importance of summarization increases with distance**   We also observed that reliance on summarization tends to increase with greater distances between the preference phrases and the final part of the prompt that would reference them. To test this, we injected irrelevant text[6] after each of the preference phrases in ToyMoodStory texts (after "John loves parties." etc.) and measured the ratio between logit difference change for the periods at the end of these phrases vs. pre-period phrases, with higher values indicating more reliance on period summaries (Table 1b). We found that the periods can be up to 15% **more** important than the actual phrases as this distance grows. Although these results are only a first step in assessing the importance of summarization importance relative to prompt length, our findings suggest that this motif may only increase in relative importance as models grow in context length, and thus merits further study.

### 4.3   VALIDATING SUMMARIZATION BEHAVIOR IN REAL-WORLD DATASETS

In order to study more rigorously how summarization behaves with natural text, we examined this phenomenon in SST (Section 2.1). We appended the suffix "Review Sentiment:" to each of the prompts and evaluate Pythia-2.8b on zero-shot classification according to whether positive or negative have higher probability and are in the top 10 tokens predicted. We then take the subset of examples that Pythia-2.8b classifies correctly that have at least one comma, which means we start with a baseline of 100% accuracy. We performed ablation and activation patching experiments (Section 2.3) on comma positions. If comma representations do not summarize sentiment information, then our experiments should not damage the model's abilities. However, our results reveal a clear summarization motif for SST.

**Ablation baselines**   We performed two baseline experiments in order to obtain a control for our later experiments. First to measure the total effect of the sentiment directions, we performed directional ablation (as described in 2.3) using the sentiment directions found with DAS to every token at every layer, resulting in a 71% reduction in the logit difference and a 38% drop in accuracy (to 62% ). Secondly, we performed directional ablation on all tokens with a small set of random directions, resulting in a $< 1\%$ change to the same metrics.

**Directional ablation at all comma positions**   We then performed directional ablation–using the DAS sentiment direction (2.2) – to every comma in each prompt, regardless of position, resulting in an 18% drop in the logit difference and an 18% drop in zero-shot classification accuracy–indicating that nearly 50% of the model's sentiment-direction-mediated ability to perform the task accurately was mediated via sentiment information at the commas. We find this particularly significant because we did not take any special effort to ensure that commas were placed at the end of sentiment phrases.

**Mean-ablation at all comma positions**   Separately from the above, we performed mean ablation at all comma positions as in 2.3, replacing each comma activation vector with the mean comma activation from the entire dataset in a layerwise fashion. Note that this changes the entire activation on the comma token, not just the activation in the sentiment direction. This resulted in a 17% drop in logit difference and an accuracy drop of 19% .

## 5   RELATED WORK

**Sentiment Analysis**   Understanding the emotional valence in text data is one of the first NLP tasks to be revolutionized by deep learning (Socher et al., 2013) and remains a popular task for benchmarking NLP models (Rosenthal et al., 2017; Nakov et al., 2016; Potts et al., 2021; Abraham et al., 2022). For a review of the literature, see (Pang & Lee, 2008; Liu, 2012; Grimes, 2014).

**Understanding Internal Representations**   This research was inspired by the field of Mechanistic Interpretability, an agenda which aims to reverse-engineer the learned algorithms inside models (Olah et al., 2020; Elhage et al., 2021b; Nanda et al., 2023a). Exploring representations (Section 3) and world-modelling behavior inside transformers has garnered significant recent interest. This

---

[6]E.g. "John loves parties. *He has a red hat and wears it everywhere, especially when he is riding his bicycle through the city streets.* Mark hates parties. *He has a purple hat but only wears it on Sundays, when he takes his weekly walk around the lake.* One day, they were invited to a grand gala. John feels very"

was studied in the context of synthetic game-playing models by Li et al. (2023) and evidence of linearity was demonstrated by Nanda (2023b) in the same context. Other work studying examples of world-modelling inside neural networks includes Li et al. (2021); Patel & Pavlick (2022); Abdou et al. (2021). Another framing of a very similar line of inquiry is the search for latent knowledge (Christiano et al., 2021; Burns et al., 2022). Prior to the transformer, representations of sentiment specifically were studied by Radford et al. (2017), notably, their finding of a sentiment neuron also implies a linear representation of sentiment.

**Causal Interventions in Language Models**   We approach our experiments from a causal mediation analysis perspective. Our approach to identifying computational subgraphs that utilize feature representations as inspired by the 'circuits analysis' framework (Stefan Heimersheim, 2023; Varma et al., 2023; Hanna et al., 2023), especially the tools of mean ablation and activation patching (Vig et al., 2020; Geiger et al., 2021; 2023a; Meng et al., 2023; Wang et al., 2022; Conmy et al., 2023; Chan et al., 2023; Cohen et al., 2023). We use Distributed Alignment Search (Geiger et al., 2023b) in order to apply these ideas to specific subspaces.

## 6   CONCLUSION

The two central novel findings of this research are the existence of a linear representation of sentiment and the use of summarization to store sentiment information. We have seen that the sentiment direction is causal and central to the circuitry of sentiment processing. Remarkably, this direction is so stark in the residual stream space that it can be found even with the most basic methods and on a tiny toy dataset, yet generalise to diverse natural language datasets from the real-world. Summarization is a motif present in larger models with longer context lengths and greater proficiency in zero-shot classification. These summaries present a tantalising glimpse into the world-modelling behavior of transformers.

**Limitations**   Many of our casual abstractions do not explain 100% of sentiment task performance. There is likely circuitry we've missed, possibly as a result of distributed representations or superposition (Elhage et al., 2022) across components and layers. This may also be a result of self-repair behavior (Wang et al., 2022; McGrath et al., 2023). Patching experiments conducted on more diverse sentence structures could also help to better isolate the circuitry for sentiment from more task-specific machinery.

The use of small datasets versus many hyperparameters and metrics poses a constant risk of gaming our own measures. Our results on the larger and more diverse SST dataset, and the consistent results across a range of models help us to be more confident in our results.

Distributed Alignment Search (DAS) outperformed on most of our metrics but presents possible dangers of overfitting to a particular dataset and taking the activations out of distribution (Lange et al., 2023). We include simpler tools such as Logistic Regression as a sanity check on our findings. Ideally, we would love to see a set of best practices to avoid such illusions.

**Implications and future work**   The summarization motif emerged naturally during our investigation of sentiment, but we would be very interested to study it in a broader range of contexts and understand what other factors of a particular model or task may influence the use of summarization.

When studying the circuitry of sentiment, we focused almost exclusively on attention heads rather than MLPs. However, early results suggest that further investigation of the role of MLPs and individual neurons is likely to yield interesting results (A.5).

Finally, we see the long-term goal of this line of research as being able to help detect dangerous computation in language models such as *deception*. Even if the existence of a single "deception direction" in activation space seems a bit naive to postulate, hopefully in the future many of the tools developed here will help to detect representations of deception or of knowledge that the model is concealing, helping to prevent possible harms from LLMs.

## REPRODUCIBILITY STATEMENT

To facilitate reproducibility of the results presented in this paper, we have provided detailed descriptions of the datasets, models, training procedures, algorithms, and analysis techniques used. We use publicly available models including GPT-2 and Pythia, with details on the specific sizes provided in Section 2.1. The methods for finding sentiment directions are described in full in Section 2.2. Our causal analysis techniques of activation patching, ablation, and directional patching are presented in Section 2.3. Circuit analysis details are extensively covered for two examples in Appendix A.3. The code for data generation, model training, and analyses will be linked in the camera-ready version of this paper.

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
