(a) PCA on adjectives in and out of sample

(b) PCA on in-sample adjectives and out-of-sample verbs

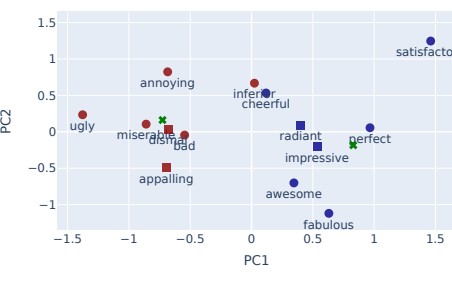

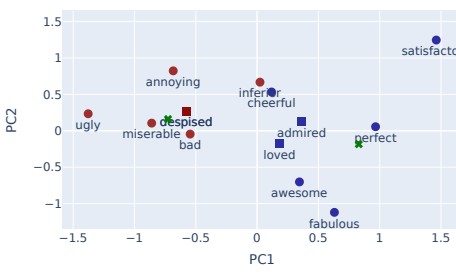

Figure A.1: 2-D PCA visualization of the embedding for a handful of adjectives and verbs (GPT2-small)

# A    APPENDIX

## A.1    FURTHER EVIDENCE FOR A LINEAR SENTIMENT REPRESENTATION

### A.1.1    CLUSTERING

In Section 2.2, we outline just a few of the many possible techniques for determining a direction which hopefully corresponds to sentiment. Is it overly optimistic to presume the existence of such a direction? The most basic requirement for such a direction to exist is that the residual stream space is clustered. We confirm this in two different ways.

First we fit 2-D PCA to the token embeddings for a set of 30 positive and 30 negative adjectives. In Figure A.1, we see that the positive adjectives (blue dots) are very well clustered compared to the negative adjectives (red dots). Moreover, we see that sentiment words which are out-of-sample with respect to the PCA (squares) also fit naturally into their appropriate color. This applies not just for unseen adjectives (Figure A.1a) but also for verbs, an entirely out-of-distribution class of word (Figure A.1b).

Secondly, we evaluate the accuracy of 2-means trained on the Simple Movie Review Continuation adjectives (Section 2.1). The fact that we can classify in-sample is not very strong evidence, but we verify that we can also classify out-of-sample with respect to the $k$-means fitting process. Indeed, even on hold-out adjectives and on the verb tokens (which are totally out of distribution), we find that the accuracy is generally very strong across models. We also evaluate on a fully out of distribution toy dataset ("simple adverbs") of the form "The traveller [adverb] walked to their destination. The traveller felt very". The results can be found in Figure A.2. This is strongly suggestive that we are stumbling on a genuine representation of sentiment.

### A.1.2    ACTIVATION ADDITION

We perform "activation addition" (Turner et al., 2023), i.e. we add a multiple of the sentiment direction to the first layer residual stream during each forward pass while generating sentence completions. We use GPT2-small for a single positive simple movie review continuation prompt: "I really enjoyed the movie, in fact I loved it. I thought the movie was just very...". We seek to verify that this can flip the generated outputs from positive to negative. The "steering coefficient" is the multiple of the sentiment direction which we add to the first layer residual stream.

By adding increasingly negative multiples of the sentiment direction, we find that indeed the completions become increasingly negative, without completely destroying the coherence of the model's generated text (Figure A.3). We are wary of taking the model's activations out of distribution using this technique, but we believe that the smoothness of the transition in combination with the

| direction | flip percent | flip median size |
|-----------|--------------|------------------|
| DAS | 96% | 107% |
| KM | 96% | 69% |
| MD | 89% | 45% |
| LR | 100% | 86% |
| PCA | 78% | 44% |

Table 2: We made a dataset of 27 negation examples and compute the change in $k$-means sentiment activation (projection of the residual stream onto the sentiment axis) at the negated token (e.g. "doubt") between the 1st and 10th resid-post layers of GPT2-small. Here "flip percent" is the percentage of the 27 prompts for which the sign of the sentiment activation has flipped and "flip median size" is the median size of the flip relative to the size of the initial sentiment activation.

knowledge of our findings in the patching setting give us some confidence that these results are meaningful.

### A.1.3 MULTI-LINGUAL SENTIMENT

We use the first few paragraphs of Harry Potter in English and French as a standard text (Elhage et al., 2021b). We find that intermediate layers of pythia-2.8b demonstrate intuitive sentiment activations for the French text (Figure A.4). It is important to note that none of the models are very good at French, but this was the smallest model where we saw hints of generalisation to other languages. The representation was not evident in the first couple of layers, probably due to the poor tokenization of French words.

### A.1.4 INTERPRETABILITY OF NEGATIONS

**Negation Flips the Sentiment Direction** Using the $k$-means sentiment direction after the first layer of GPT2-small, we can obtain a view of how the model updates its view of sentiment during the forward pass, analogous to the "logit lens" technique from nostalgebraist (2020). The example text that we use here is "You never fail. Don't doubt it. I am not uncertain". In Figure A.5, we see how the sentiment activation flips when the context of the sentiment word denotes that it is negated. Words like 'fail', 'doubt' and 'uncertain' can be seen to flip from negative in the first couple of layers to being positive after a few layers of processing.

We quantified this result using a toy dataset of 27 similar examples and computed the flip in sentiment activation during the forward pass for different direction finding methods (Table 2).

An interesting task for future circuits analysis research could be to better understand the circuitry used to flip the sentiment axis in the presence of a negation context. We suspect significant MLP involvement (see Section A.5).

### A.1.5 UNIVERSALITY EXAMPLES

For comparison with Figure 2, we include Figure A.6, where we visualise the similarity and classification accuracy of directions found by different methods, this time for GPT2-small (Section 2.1) and StableLM 3B (Tow) instead of pythia-2.8b.

### A.2 LIMITATIONS TO OUR LINEARITY CLAIM

Did we find a truly universal sentiment direction, or merely the first principal component of directions used across different sentiment tasks? As found by Bricken et al. (2023), we suspect that this feature could be "split" further into more specific sentiment features. We performed an experiment to help validate that the common sentiment feature across tasks is one dimensional. DAS can be used not just to find a causally impactful direction, but a causal subspace of any dimension. Figure A.7 demonstrates that whilst increasing the DAS dimension improves the patching metric in-sample (A.7a), the metric does not improve out-of-distribution (A.7b).

Similarly, one might wonder if there is really a single bipolar sentiment direction or if we have simply found the difference between a "positive" and a "negative" sentiment direction. It turns out that this distinction is not well-defined, given that we find empirically that there is a direction corresponding to "valenced words". Indeed, if $x$ is the valence direction and $y$ is the sentiment direction, then $p = x + y$ represents positive sentiment and $n = x - y$ is the negative direction. Conversely, we can reframe as starting from the positive/negative directions $p$ and $n$, and then re-derive $x = \frac{p+n}{2}$ and $y := \frac{p-n}{2}$.

### A.3 DETAILED CIRCUIT ANALYSIS

In order to build a picture of each circuit, we used the process pioneered in Wang et al. (2022):

- Identify which model components have the greatest impact on the logit difference when path patching is applied (with the final result of the residual stream set as the receiver).
- Examine the attention patterns (value-weighted, in some cases) and other behaviors of these components (in practice, attention heads) in order to get a rough idea of what function they are performing.
- Perform path-patching using these heads (or a distinct cluster of them) as receivers.
- Repeat the process recursively, performing contextual analyses of each "level" of attention heads in order to understand what they are doing, and continuing to trace the circuit backwards.

In each path-patching experiment, change in logit difference is used as the patching metric. We started with GPT-2 as an example of a classic LLM displays a wide range of behaviors of interest, and moved to larger models when necessary for the task we wanted to study (choosing, in each case, the smallest model that could do the task).

#### A.3.1 SIMPLE SENTIMENT - GPT-2 SMALL

We examined the circuit performing tasks for the following sentence template:

> I thought this movie was ADJECTIVE, I VERBed it. Conclusion: This movie is

Using a threshold of 5%-or-greater damage to the logit difference for our patching experiments, we found that GPT-2 Small contained 4 primary heads contributing to the most proximate level of circuit function–10.4, 9.2, 10.1, and 8.5 (using "layer.head" notation). Examining their value-weighted attention patterns, we found that attention to ADJ and VRB in the sentence was most prominent in the first three heads, but 8.5 attended primarily to the second "movie" token. We also observed that 9.2 attended to this token as well as to ADJ. (Results of activation patching can be seen in Fig. A.8.)

Conducting path-patching with 8.5 and 9.2 as receivers, we identified two heads–7.1 and 7.5–that primarily attend to ADJ and VRB from the "movie" token. We further determined that the output of these heads, when path-patched through 9.2 and 8.5 as receivers, was causally important to the circuit (with patching causing a logit difference shift of 7% and 4% respectively for 7.1 and 7.5). This was not the case for other token positions, which demonstrates that causally relevant information is indeed being specially written to the "movie" position. We thus designated it the SUM token in this circuit, and we label 8.5 a summary-reader head.

Repeating our analysis with lower thresholds yielded more heads with the same behavior but weaker effect sizes, adding 9.10, 11.9, and 6.4 as summary reader, direct sentiment reader, and sentiment summarizer respectively. This gives a total of 9 heads making up the circuit.

#### A.3.2 TOYMOODSTORY CIRCUIT - PYTHIA-2.8B

We also examined the circuit for this sentence template: Carl hates parties, and avoids them whenever possible. Jack loves parties, and joins them whenever possible. One day, they were invited to a grand gala. Jack feels very [excited/nervous]. We did not attempt to reverse-engineer the entire circuit, but examined it from the perspective of what matters causally for sentiment processing–especially determining to what extent summarization occurred.

Following the same process as with GPT-2 with preference/sentiment-flipped prompts (that is, taking $x_{orig}$ to be "John hates parties,... Mary loves parties," and $x_{flipped}$ to be "John loves parties,...

Mary hates parties"), we initially identified 5 key heads that were most causally important to the logit difference at END: 17.19, 22.5, 14.4, 20.10, and 12.2 (in "layer.head" notation). Examining the value-weighted attention patterns, we observed that the top token receiving attention from END was always the repeated name RNAME (e.g., "John" in "John feels very") or the "feels" token FEEL, indicating that some summarization may have taken place there.

We also observed that the top token attended to from RNAME and FEEL was in fact the comma at the end of the queried preference phrase (that is, the comma at the end of "John hates parties"). We designate this position COMMASUM.

**Multi-functional heads**    Interestingly, we observed that most of these heads were multi-functional: that is, they both attended to COMMASUM from RNAME and FEEL, and also attended to RNAME and FEEL from END, producing output in the direction of the logit difference. This is possible because these heads exist at different layers, and later heads can read the summarized information from previous heads as well as writing their own summary information.

**Direct effect heads**    Specifically, the direct effect heads were:

- Head 17.19 did not attend to commas significantly, but did attend to the periods at the end of each preference sentence in addition to its primary attention to RNAME and FEEL, and did not display COMMASUM-reading behavior.
- Head 22.5 attended almost exclusively to FEEL, and did not display COMMASUM-reading behavior.
- Other direct effect heads (14.4, 20.10 and 12.2) did show COMMASUM-reading behavior as well as reading from the near-end tokens to produce output in the direction of the logit difference. In each case, we verified with path-patching that information from these positions was causally relevant.

**Name summary writers**    We also found important heads (12.17 being by far the most important) that are only engaged with attending to COMMASUM and producing output at RNAME and FEEL.

**Comma summary writers**    We further investigated what circuitry was causally important to task performance mediated through the COMMASUM positions, but did not flesh this out in full detail; after finding initial examples of summarization, we focused on its causal relevance and interaction with the sentiment direction, leaving deeper investigation to future work.

### A.4    ADDITIONAL SUMMARIZATION FINDINGS

#### A.4.1    CIRCUITRY FOR PROCESSING COMMAS VS. ORIGINAL PHRASES IS SEMI-SEPARATE

Though there is overlap between the attention heads involved in the circuitry for processing sentiment from key phrases and that from summarization points, there are also some clear differences, suggesting that the ability to read summaries could be a specific capability developed by the model (rather than the model simply attending to high-sentiment tokens).

As can be seen in Figure A.9, there are distinct groups of attention heads that result in damage to the logit difference in different situations–that is, some react when phrases are patched, some react disproportionately to comma patching, and one head seems to have a strong response for either patching case. This is suggestive of semi-separate summary-reading circuitry, and we hope future work will result in further insights in this direction.

#### A.4.2    RESULTS FROM OTHER MODELS

We replicated the ToyMoodStories comma-swapping experiment (as explained in Section 4.2) in Pythia-6.9b and Mistral-7b, resulting in the findings shown in Table 3.

We take this as evidence that the comma-summarization phenomenon is not limited exclusively to Pythia-2.8b.

| Intervention | Pythia-2.8b | Pythia-6.9b | Mistral-7b |
|---|---|---|---|
| Patching full phrase values (incl. commas) | -75% | -152% | -155% |
| Patching pre-comma values (freezing commas & periods) | -38% | -46% | -16% |
| Patching comma and period values only | -37% | -68% | -100% |

Table 3: Change in logit difference from patching at commas in ToyMoodStory in three different models

## A.5 Neurons writing to sentiment direction in GPT2-small are interpretable

We observed that the cosine similarities of neuron out-directions with the sentiment direction are extremely heavy tailed (Figure A.11). Thanks to Neuroscope (Nanda, 2023a), we can quickly see whether these neurons are interpretable. Indeed, here are a few examples from the tails of that distribution:

- L3N1605 activates on "hesitate" following a negation

- Neuron L6N828 seems to be activating on words like "however" or "on the other hand" *if* they follow something negative

- Neuron L5N671 activates on negative words that follow a "not" contraction (e.g. didn't, doesn't)

- L6N1237 activates strongly on "but" following "not bad"

We take L3N1605, the "not hesitate" neuron, as an extended example and trace backwards through the network using Direct Logit Attribution[7]. We computed the relative effect of different model components on L3N1605 in the two different cases "I would not hesitate" vs. "I would always hesitate". The main contributors to this difference are L1H0, L3H10, L3H11 and MLP2. Expanding out MLP2 into individual neurons we find that the contributions to L3N1605 are sparse. For example, L2N1154 activates on words like "don't", "not", "no", etc. It activates on "not" but not "hesitate" in "I would not hesitate" but activates on "hesitate" in "I would always hesitate". Visualizing the attention pattern of L1H0 shows that it attends from "hesitate" to the previous token if it is "not", but not if it is "always".

These anecdotal examples suggest at a complex network of machinery for transmitting sentiment information across components of the network using a single critical axis of the residual stream as a communication channel. We think that exploring these neurons further could be a very interesting avenue of future research, particularly for understanding how the model updates sentiment based on negations where these neurons seem to play a critical role.

## A.6 Detailed description of metrics

- **Logit Difference:** We extend the logit difference metric used by Wang et al. (2022) to the setting with 2 *classes* of next token rather than only 2 valid next tokens. This is useful in situations where there are many possible choices of positively or negatively valenced next tokens. Specifically, we examine the average difference in logits between sets of positive/negative next-tokens $T^{\text{positive}} = \{t_i^{\text{positive}} : 1 \leq i \leq n\}$ and $T^{\text{negative}} = \{t_i^{\text{negative}} : 1 \leq i \leq n\}$ in order to get a smooth measure of the model's ability to differentiate between sentiment. That is, we define the logit difference as $\frac{1}{n} \sum_i \left[ \text{logit}(t_i^{\text{positive}}) - \text{logit}(t_i^{\text{negative}}) \right]$. Larger differences indicate more robust separation of the positive/negative tokens, and zero or inverted differences indicate zero or inverted sentiment processing respectively. When

---

[7]This technique decomposes model outputs into the sum of contributions of each component, using the insight from Elhage et al. (2021b) that components are independent and additive

used as a patching metric, this demonstrates the causal efficacy of various interventions like activation patching or ablation.[8]

- **Logit Flip:** Similar to logit difference, this is the percentage of cases where the logit difference between $T^{\text{positive}}$ and $T^{\text{negative}}$ is inverted after a causal intervention. This is a more discrete measure which is helpful for gauging whether the magnitude of the logit differences is sufficient to actually flip model predictions.

- **Accuracy:** Out of a set of prompts, the percentage for which the logits for tokens $T^{\text{correct}}$ are greater than $T^{\text{incorrect}}$. In practice, usually each of these sets only has one member (e.g., "Positive" and "Negative").

## A.7 GLOSSARY

### GLOSSARY

**ablation** A technique where we eliminate the contribution of a particular component to a model's output (usually by replacing the component's output with zeros or the mean over some dataset or a random sample from some dataset) in order to demonstrate the magnitude of its importance.

**activation addition** Formerly called "activation steering", a technique from Turner et al. (2023) where a vector is added to the residual stream at a certain position (or all positions) and layer during each forward pass while generating sentence completions. In our case, the vector is the sentiment direction.

**activation patching** A technique introduced in Meng et al. (2023), under the name 'causal tracing', which uses a causal intervention to identify which activations in a model matter for producing some output. It runs the model on some 'clean' input, replaces (patches) an activation with that same activation on 'flipped' input, and sees how much that shifts the output from 'clean' to 'flipped'.

**activation steering** See activation addition.

**circuit** A computational subgraph of a neural network which performs some human-interpretable task (Wang et al., 2022).

**DAS** Distributed Alignment Search (Geiger et al., 2023b) uses gradient descent to train a rotation matrix representing an orthonormal change of basis to one better aligned with the model's features. We mostly focus on a special case of finding a singular critical direction, where we patch along the first dimension of the rotated basis and then use a smooth patching metric (such as the logit difference between positive and negative completions) as the objective to be minimised.

**direct effect head** An attention head writes directly to the final token logits, affecting the output on a specific task.

**directional ablation** A form of ablation experiment in which restrict the intervention to a single dimension. That is, assuming mean ablation, for dimension $d$ and prompt index $i$ out of $n$, we replace the residual stream vector $r_i$ with $r_i - r_i \cdot d + \sum_j \frac{r_j \cdot d}{n}$.

**directional activation patching** A variant of activation patching introduced in this paper where we only patch a single dimension from a counterfactual activation. That is, for prompts $x_{\text{orig}}$ and $x_{\text{new}}$, direction $d$, a set of model components $\mathbb{C}$, we run a forward pass on $x_{\text{orig}}$ but for each component in $\mathbb{C}$, we patch/replace the output $o_{\text{orig}}$ with $o_{\text{orig}} - o_{\text{orig}} \cdot d + o_{\text{new}} \cdot d$. This is equivalent to activation patching a single neuron, but done in a rotated basis (where $d$ is the first column of the rotation matrix).

---

[8]We use this metric often because it is more sensitive than accuracy to small shifts in model behavior, which is particularly useful for circuit identification where the effect size is small but real. That is, in many cases a token of interest might become much more likely but not cross the threshold to change accuracy metrics, and in this case logit difference will detect it. Logit difference is also useful when trying to measure the model behavior transition between two different, opposing prompts–in this case, the logit difference for each of the prompts is used for lower and upper baselines, and we can measure the degree to which the logit difference behavior moves from one pole to the other.

**directional patching**  See directional activation patching.

**mean ablation**  A type of ablation method, where we seek to eliminate the contribution of a particular component to demonstrate its importance, where we replace a particular set of activations with their mean over an appropriate dataset.

**patching metric**  A summary statistic used to quantify the results of an activation patching experiment. By default here we use the percentage change in logit difference as in Wang et al. (2022).

**path patching**  A variant of activation patching introduced in Wang et al. (2022) in which only the activations related to the residual stream paths between two sets of endpoints (senders and receivers) are patched, but the remainder of the network upstream of the receivers is frozen. Given a set $R$ of receivers, a sender attention head $h$, and paths $P$ between $h$ and each of $R$, activations from the mirrored dataset are patched into $P$ while keeping the remainder of the network fixed (aside from everything downstream of $R$).

**sentiment activation**  The projection of the residual stream at a given token position and layer onto the sentiment direction.

**sentiment direction**  The direction in the residual stream space associated with the sentiment feature.

**sentiment summarizer**  An attention head which is a critical component of a sentiment-driven task and acts via V-composition, writing information to an intermediate token position which is later read by a direct effect head.

**SST**  Stanford Sentiment Treebank is a labelled sentiment dataset from Socher et al. (2013) described in Section 2.1.

**summarization motif**  The phenomenon where sentiment is not solely represented on emotionally charged words, but is additionally summarised at intermediate positions without inherent sentiment, such as punctuation and names.

**V-composition**  When the value vectors of a downstream head contain information written by the output of an upstream attention head (Elhage et al., 2021b).

**value-weighted attention**  The attention pattern weighted by the norm of the value vector at each position as per Kobayashi et al. (2020). We favor this over the raw attention pattern as it filters for *significant* information being moved.

**zero ablation**  A type of ablation method, where we seek to eliminate the contribution of a particular component to demonstrate its importance, where we replace a particular set of activations with their mean over an appropriate dataset.

kmeans accuracy (gpt2-small)

| train_set | train_pos | train_layer | test_set | simple_test | | simple_adve |
|---|---|---|---|---|---|---|
| | | | test_pos | ADJ | VRB | ADV |
| simple_train | ADJ | 0 | | 100.0% | 83.3% | 50.0% |
| | | 1 | | 100.0% | 100.0% | 55.3% |
| | | 2 | | 100.0% | 100.0% | 60.5% |
| | | 3 | | 100.0% | 100.0% | 65.8% |
| | | 4 | | 100.0% | 100.0% | 78.9% |
| | | 5 | | 100.0% | 100.0% | 57.9% |
| | | 6 | | 100.0% | 100.0% | 84.2% |
| | | 7 | | 100.0% | 100.0% | 71.1% |
| | | 8 | | 100.0% | 100.0% | 65.8% |
| | | 9 | | 100.0% | 100.0% | 68.4% |
| | | 10 | | 91.7% | 100.0% | 60.5% |
| | | 11 | | 91.7% | 100.0% | 60.5% |
| | | 12 | | 33.3% | 58.3% | 31.6% |

(a) GPT-2 Small

kmeans accuracy (gpt2-medium)

| train_set | train_pos | train_layer | test_set | simple_test | | simple_adverb |
|---|---|---|---|---|---|---|
| | | | test_pos | ADJ | VRB | ADV |
| simple_train | ADJ | 0 | | 100.0% | 100.0% | 50.0% |
| | | 1 | | 100.0% | 83.3% | 50.0% |
| | | 2 | | 100.0% | 100.0% | 47.4% |
| | | 3 | | 91.7% | 100.0% | 47.4% |
| | | 4 | | 91.7% | 100.0% | 47.4% |
| | | 5 | | 100.0% | 100.0% | 47.4% |
| | | 6 | | 100.0% | 100.0% | 68.4% |
| | | 7 | | 91.7% | 100.0% | 50.0% |
| | | 8 | | 91.7% | 100.0% | 84.2% |
| | | 9 | | 100.0% | 100.0% | 86.8% |
| | | 10 | | 100.0% | 100.0% | 71.1% |
| | | 11 | | 100.0% | 100.0% | 94.7% |
| | | 12 | | 100.0% | 100.0% | 65.8% |
| | | 13 | | 100.0% | 100.0% | 63.2% |
| | | 14 | | 100.0% | 100.0% | 73.7% |
| | | 15 | | 100.0% | 100.0% | 60.5% |
| | | 16 | | 100.0% | 100.0% | 57.9% |
| | | 17 | | 100.0% | 100.0% | 55.3% |
| | | 18 | | 100.0% | 100.0% | 55.3% |
| | | 19 | | 100.0% | 100.0% | 76.3% |
| | | 20 | | 100.0% | 100.0% | 84.2% |
| | | 21 | | 100.0% | 91.7% | 65.8% |
| | | 22 | | 100.0% | 100.0% | 52.6% |
| | | 23 | | 100.0% | 100.0% | 57.9% |
| | | 24 | | 83.3% | 58.3% | 50.0% |

(b) GPT-2 Medium

kmeans accuracy (gpt2-large)

| train_set | train_pos | train_layer | test_set | simple_test | | simple_adverb |
|---|---|---|---|---|---|---|
| | | | test_pos | ADJ | VRB | ADV |
| simple_train | ADJ | 0 | | 100.0% | 100.0% | 47.4% |
| | | 1 | | 100.0% | 100.0% | 42.1% |
| | | 2 | | 91.7% | 100.0% | 47.4% |
| | | 3 | | 100.0% | 100.0% | 50.0% |
| | | 4 | | 100.0% | 100.0% | 50.0% |
| | | 5 | | 100.0% | 100.0% | 71.1% |
| | | 6 | | 100.0% | 100.0% | 55.3% |
| | | 7 | | 100.0% | 100.0% | 78.9% |
| | | 8 | | 100.0% | 100.0% | 76.3% |
| | | 9 | | 100.0% | 100.0% | 78.9% |
| | | 10 | | 100.0% | 100.0% | 81.6% |
| | | 11 | | 100.0% | 100.0% | 86.8% |
| | | 12 | | 100.0% | 100.0% | 86.8% |
| | | 13 | | 100.0% | 100.0% | 86.8% |
| | | 14 | | 100.0% | 100.0% | 78.9% |
| | | 15 | | 100.0% | 100.0% | 68.4% |
| | | 16 | | 100.0% | 100.0% | 68.4% |
| | | 17 | | 100.0% | 100.0% | 71.1% |
| | | 18 | | 100.0% | 100.0% | 78.9% |
| | | 19 | | 100.0% | 100.0% | 84.2% |
| | | 20 | | 100.0% | 100.0% | 73.7% |
| | | 21 | | 100.0% | 100.0% | 71.1% |
| | | 22 | | 100.0% | 100.0% | 60.5% |
| | | 23 | | 100.0% | 100.0% | 52.6% |
| | | 24 | | 100.0% | 100.0% | 50.0% |

(c) GPT-2 Large

kmeans accuracy (gpt2-xl)

| train_set | train_pos | train_layer | test_set | simple_test | | simple_adverb |
|---|---|---|---|---|---|---|
| | | | test_pos | ADJ | VRB | ADV |
| simple_train | ADJ | 0 | | 100.0% | 100.0% | 52.6% |
| | | 1 | | 91.7% | 100.0% | 50.0% |
| | | 2 | | 100.0% | 100.0% | 50.0% |
| | | 3 | | 100.0% | 83.3% | 50.0% |
| | | 4 | | 100.0% | 100.0% | 50.0% |
| | | 5 | | 100.0% | 100.0% | 50.0% |
| | | 6 | | 100.0% | 100.0% | 47.4% |
| | | 7 | | 100.0% | 100.0% | 44.7% |
| | | 8 | | 100.0% | 100.0% | 44.7% |
| | | 9 | | 100.0% | 100.0% | 44.7% |
| | | 10 | | 100.0% | 100.0% | 55.3% |
| | | 11 | | 100.0% | 100.0% | 52.6% |
| | | 12 | | 100.0% | 100.0% | 63.2% |
| | | 13 | | 100.0% | 100.0% | 63.2% |
| | | 14 | | 100.0% | 100.0% | 81.6% |
| | | 15 | | 100.0% | 100.0% | 63.2% |
| | | 16 | | 100.0% | 100.0% | 57.9% |
| | | 17 | | 100.0% | 100.0% | 94.7% |
| | | 18 | | 100.0% | 100.0% | 60.5% |
| | | 19 | | 100.0% | 100.0% | 81.6% |
| | | 20 | | 100.0% | 100.0% | 89.5% |
| | | 21 | | 100.0% | 100.0% | 86.8% |
| | | 22 | | 100.0% | 100.0% | 89.5% |
| | | 23 | | 100.0% | 100.0% | 86.8% |
| | | 24 | | 100.0% | 100.0% | 89.5% |

(d) GPT-2 XL

Figure A.2: 2-means classification accuracy for various GPT-2 sizes, split by layer (showing up to 24 layers)

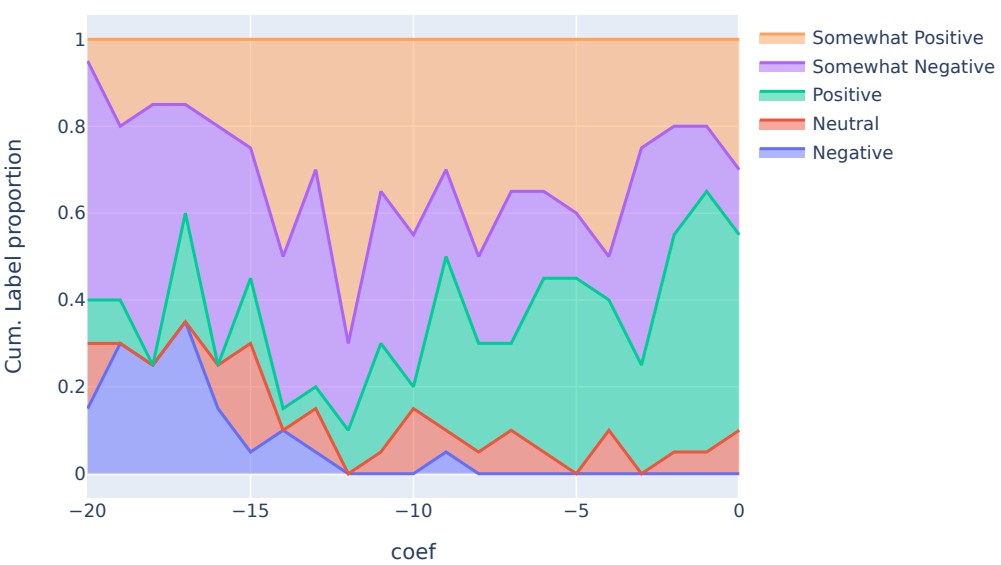

Figure A.3: Area plot of sentiment labels for generated outputs by activation steering coefficient, starting from a single positive movie review continuation prompt. Activation addition (Turner et al., 2023) was performed in GPT2-small's first residual stream layer. Classification was performed by GPT-4.

<|endoftext|>

Mr. and Mrs. Dursley, of number four, Privet Drive, were proud to say that they were perfectly normal, thank you very much. They were the last people you'd expect to be involved in anything strange or mysterious, because they just didn't hold with such nonsense.

Mr. Dursley was the director of a firm called Grunnings, which made drills. He was a big, beefy man with hardly any neck, although he did have a very large mustache. Mrs. Dursley was thin and blonde and had nearly twice the usual amount of neck, which came in very useful as she spent so much of her time craning over garden fences, spying on the neighbors. The Dursleys had a small son called Dudley and in their opinion there was no finer boy anywhere.

The Dursleys had everything they wanted, but they also had a secret, and their greatest fear was that somebody would discover it. They didn't think they could bear it if anyone found out about the Potters. Mrs. Potter was Mrs. Dursley's sister, but they hadn't met for several years; in fact, Mrs. Dursley pretended she didn't have a sister, because her sister and her good-for-nothing husband were as unDursleyish as it was possible to be. The Dursleys shuddered to think what the neighbors would say if the Potters arrived in the street. The Dursleys knew that the Potters had a small son, too, but they had never even seen him. This boy was another good reason for keeping the Potters away; they didn't want Dudley mixing with a child like that.

When Mr. and Mrs. Dursley woke up on the dull, gray Tuesday our story starts, there was nothing about the cloudy sky outside to suggest that strange and mysterious things would soon be happening all over the country. Mr. Dursley hummed as he picked out his most boring tie for work, and Mrs. Dursley gossiped away happily as she wrestled a screaming Dudley into his high chair.

(a) First 4 paragraphs of Harry Potter in English

<|endoftext|>

Mr et Mrs Dursley, qui habitaient au 4, Privet Drive, avaient toujours affirmé avec la plus grande fierté qu'ils étaient parfaitement normaux, merci pour eux. Jamais quiconque n'aurait imaginé qu'ils puissent se trouver impliqués dans quoi que ce soit d'étrange ou de mystérieux. Ils n'avaient pas de temps à perdre avec des sornettes.

Mr Dursley dirigeait la Grunnings, une entreprise qui fabriquait des perceuses. C'était un homme grand et massif, qui n'avait pratiquement pas de cou, mais possédait en revanche une moustache de belle taille. Mrs Dursley, quant à elle, était mince et blonde et disposait d'un cou deux fois plus long que la moyenne, ce qui lui était fort utile pour espionner ses voisins en regardant par-dessus les clôtures des jardins. Les Dursley avaient un petit garçon prénommé Dudley et c'était à leurs yeux le plus bel enfant du monde.

Les Dursley avaient tout ce qu'ils voulaient. La seule chose indésirable qu'ils possédaient, c'était un secret dont ils craignaient plus que tout qu'on le découvre un jour. Si jamais quiconque venait à entendre parler des Potter, ils étaient convaincus qu'ils ne s'en remettraient pas. Mrs Potter était la soeur de Mrs Dursley, mais toutes deux ne s'étaient plus revues depuis des années. En fait, Mrs Dursley faisait comme si elle était fille unique, car sa soeur et son bon à rien de mari étaient aussi éloignés que possible de tout ce qui faisait un Dursley. Les Dursley tremblaient d'épouvante à la pensée de ce que diraient les voisins si par malheur les Potter se montraient dans leur rue. Ils savaient que les Potter, eux aussi, avaient un petit garçon, mais ils ne l'avaient jamais vu. Son existence constituait une raison supplémentaire de tenir les Potter à distance : il n'était pas question que le petit Dudley se mette à fréquenter un enfant comme celui-là.

Lorsque Mr et Mrs Dursley s'éveillèrent, au matin du mardi où commence cette histoire, il faisait gris et triste et rien dans le ciel nuageux ne laissait prévoir que des choses étranges et mystérieuses allaient bientôt se produire dans tout le pays. Mr Dursley fredonnait un air en nouant sa cravate la plus sinistre pour aller travailler et Mrs Dursley racontait d'un ton badin les derniers potins du quartier en s'efforçant d'installer sur sa chaise de bébé le jeune Dudley qui braillait de toute la force de ses poumons.

(b) First 3 paragraphs of Harry Potter in French

23

Figure A.4: First paragraphs of Harry Potter in different languages. Model: pythia-2.8b.

Figure A.5: Visualizing the sentiment activations across layers for a text where the sentiment hinges on negations. Color represents sentiment activation (projection of the residual stream onto the sentiment axis)) at the given layer and position. Red is negative, blue is positive. Each row is a residual stream layer, first layer is at the top. Model: GPT2-small

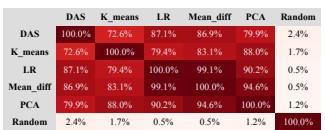

(a) Cosine similarity of directions learned by different methods in **GPT2-small**'s first layer. Each sentiment direction was derived from *adjective* representations in the ToyMovieReview dataset (Section 2.1).

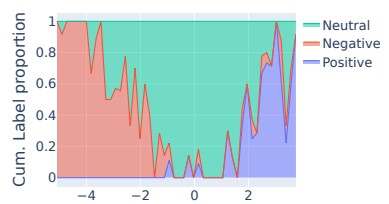

(b) Area plot of sentiment labels for OpenWebText samples by sentiment activation, i.e. the projection of the first residual stream layer of **GPT2-small** at that token onto the sentiment direction (left). Accuracy using sentiment activations to classify tokens as positive or negative (right). The threshold taken is the top/bottom 0.1% of activations over OpenWebText. Classification was performed by GPT-4.

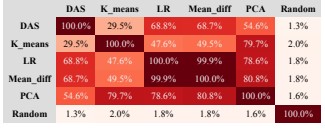

(c) Cosine similarity of directions learned by different methods in **Stable LM**'s first layer. Each sentiment direction was derived from *adjective* representations in the ToyMovieReview dataset (Section 2.1).

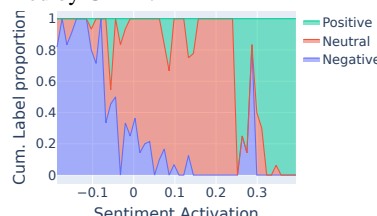

(d) Area plot of sentiment labels for OpenWebText samples by sentiment activation, i.e. the projection of the first residual stream layer of **Stable LM 3B** at that token onto the sentiment direction (left). Accuracy using sentiment activations to classify tokens as positive or negative (right). The threshold taken is the top/bottom 0.1% of activations over OpenWebText. Classification was performed by GPT-4.

Figure A.6: A study in universality: correlational analysis of sentiment directions in GPT2-small (Section 2.1) and Stable LM 3B (Tow) instead of pythia-2.8b, cf. Figure 2.

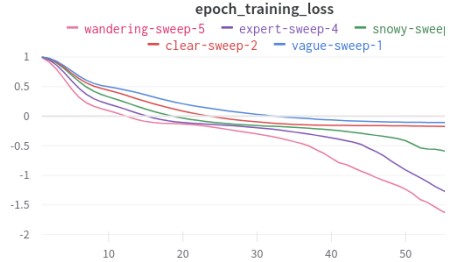

(a) Training loss for DAS on adjectives in a toy movie review dataset

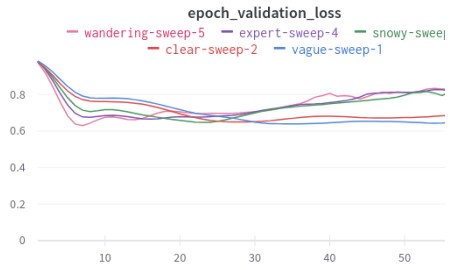

(b) Validation loss for DAS on a simple character mood dataset with a varying adverb

Figure A.7: DAS sweep over the subspace dimension (GPT2-small). The runs are labelled with the integer $n$ where $d_{\text{DAS}} = 2^{n-1}$. Loss is 1 minus the usual patching metric.

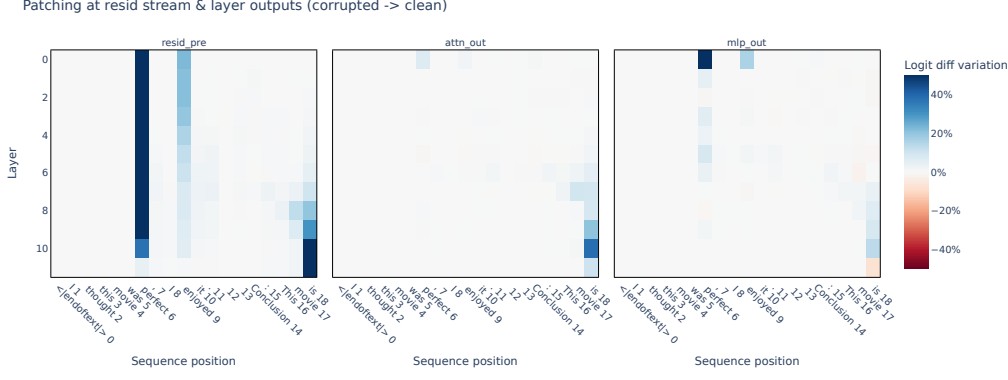

Figure A.8: Activation patching results for the GPT-2 Small ToyMovieReview circuit, showing how much of the original logit difference is recaptured when swapping in activations from $x_{orig}$ (when the model is otherwise run on $x_{flipped}$). Note that attention output is only important at the SUM position, and that this information is important to task performance at the residual stream layers (8 and 9) in which the summary-readers reside. Other than this, the most important residual stream information lies at the ADJ and VRB positions.

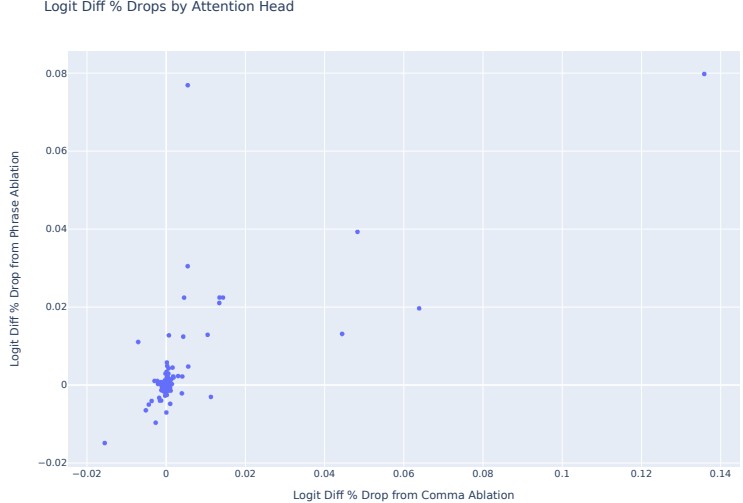

Figure A.9: Logit difference drops by head when commas or pre-comma phrases are patched. Model: pythia-2.8b.

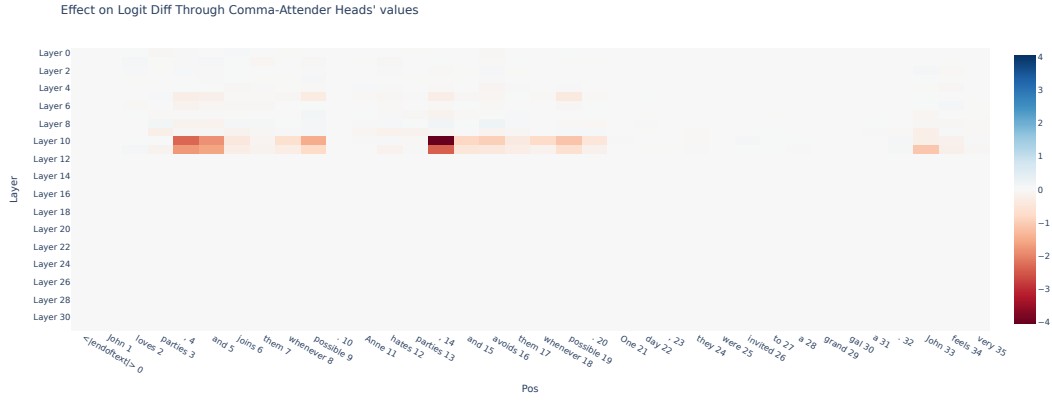

Figure A.10: Path-patching commas and comma phrases in Pythia-2.8b, with attention heads L12H2 and L12H17 writing to repeated name and "feels" as receivers. Patching the paths between the comma positions and the receiver heads results in the greatest performance drop for these heads.

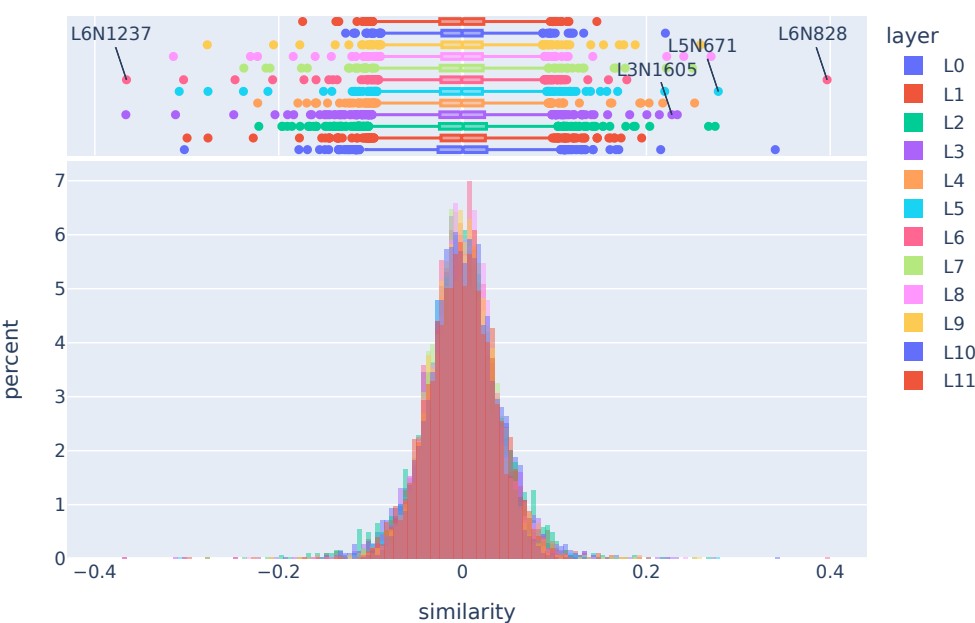

Figure A.11: Cosine similarity of neuron out-directions and the sentiment direction in GPT2-small