# OpenReview forum: "Language Models Linearly Represent Sentiment"
_ICLR.cc/2024/Conference — Submitted to ICLR 2024_

### Official Review · Reviewer_6Ynr · 2023-10-31

**Soundness:** 3 good
**Presentation:** 2 fair
**Contribution:** 2 fair
**Rating:** 6
**Confidence:** 3

**Summary:**

In this work, the authors delve into the representation of sentiment in Language Models (LLMs) and summarize two pivotal discoveries: the existence of a linear sentiment representation and the utilization of summarization for storing sentiment information. Regarding the first finding, this paper isolates the activation direction and demonstrates its causal relevance in both toy tasks and real-world datasets. As for the second discovery, this study uncovers the underlying mechanisms associated with the aforementioned direction, emphasizing the roles of a select group of attention heads and neurons. To investigate and corroborate these conclusions, the paper conducts experiments using four datasets, as well as GPT-2 and Pythia models.

**Strengths:**

1. The content explored in this paper is intriguing, as it delves into the representation of sentiment in LLMs, a variable in the data generation process that pertains to various language tasks. This research direction contributes to identifying potential risks in language models, including deception and the concealment of model knowledge, thereby mitigating potential harm caused by LLMs.

2. The paper exhibits a well-structured format with lucid expression, comprehensive experimental exploration, and a thorough examination of limitations, implications, and future work in the concluding section.

**Weaknesses:**

1. The statements in the abstract and introduction indicate that this paper investigates the representation of sentiment in LLMs. However, the experiments employ GPT2-small for movie review continuation, Pythia-1.4b for classification, and Pythia-2.8b for multi-subject tasks. GPT and Pythia may not be representative of the broader spectrum of LLMs, which reduces the persuasiveness of the conclusions drawn from the study.

2. There is no caption provided for Table 1 in the paper. Furthermore, Figure 2 is predominantly featured in Section 3, but there exists a substantial gap between its initial reference and the section in which it is discussed.

3. The proposed ToyMovieReview format, "I thought this movie was ADJECTIVE, I VERBed it. Conclusion: This movie is," appears to be more suitable for sentence-level sentiment analysis. It may not be as effective for other sentiment analysis tasks, such as document-level sentiment analysis or fine-grained sentiment analysis. Furthermore, the rationale for emphasizing adjectives and verbs in this specific prompt design remains unclear, as it has not been adequately explained in previous sections of the paper.

4. In Section 4.1, the authors find that the model employs a simple and interpretable algorithm to perform the task. However, this algorithm has not been supported by corresponding experiments, thus failing to demonstrate and illustrate the use of summarization for retaining sentiment information.

**Questions:**

Pleas see the Weaknesses.

---

> ### Author Response · Authors · 2023-11-17
>
> We greatly appreciate your constructive review and the positive remarks on our manuscript's structure, clarity, and experimental design. We particularly appreciate your interest in the relevance of our findings for mitigating the potential risks from language models.
>
> > The statements in the abstract and introduction indicate that this paper investigates the representation of sentiment in LLMs… GPT and Pythia may not be representative of the broader spectrum of LLMs, which reduces the persuasiveness of the conclusions drawn from the study.
>
> We acknowledge your concern about the representativeness of our findings across the broader spectrum of large language models (LLMs). In our latest revision, we have repeated the experiments in Figures 2a and 2b (i.e. demonstrating the consistency and classification accuracy of the sentiment direction) for Pythia-2.8b to compare (in Section A.1.5) with GPT2-small (now in Figure A.6). We are also conducting additional experiments across more model families to demonstrate the summarization motif and the linear representation of sentiment. Our in-progress experiments include a comparison with two StableLM models up to 7B parameters in size which are trained on significantly more data than Pythia, making them closer in capability to today’s state of the art models.
>
> However, we would like to note that in our work so far, our patching experiments are shown for different sizes of Pythia and for GPT2-small in Figure 2e, demonstrating our core claim that across a range of models, there exists a linear and causally relevant sentiment direction. Moreover, in Appendix Figure A.2, we show that for different sizes of GPT2, k-means can classify tokens out of distribution. The largest model studied was 35 times larger than the smallest and the two distinct families of models here are quite different. The families have different training corpuses (Pythia was trained with extensive code and ArXiv papers, unlike GPT-2) and different training and architectural choices (Pythia uses rotary positional encodings, untied embeddings, and was not trained with dropout). These differences make the existence of a similar direction in the residual stream particularly significant.
>
> Also, as noted in the paper, unfortunately some models were too small to accomplish the relevant task, rendering a comparison impossible. On the other end of the spectrum, working with large models presented significant computational challenges.
>
> > There is no caption provided for Table 1 in the paper. Furthermore, Figure 2 is predominantly featured in Section 3, but there exists a substantial gap between its initial reference and the section in which it is discussed.
>
> Thank you very much for pointing out these slips in presentation. In the revised version we have ensured that all tables and figures are appropriately captioned and, where possible, placed in proximity to the relevant text for ease of reference and to facilitate a coherent narrative flow.
>
> > The proposed ToyMovieReview format, "I thought this movie was ADJECTIVE, I VERBed it. Conclusion: This movie is," appears to be more suitable for sentence-level sentiment analysis… Furthermore, the rationale for emphasizing adjectives and verbs in this specific prompt design remains unclear, as it has not been adequately explained in previous sections of the paper.
>
> Your point about the ToyMovieReview dataset is well-taken. To address this, we provide the additional experiment of replicating the SST all-token ablation experiment on the IMDB reviews dataset and have added the results to Section 3.3. These prompts are significantly longer and thus the results better demonstrate our findings at the document level.
>
> We realize that our choice to focus on adjectives and verbs in the ToyMovieReview dataset was not fully elucidated. This choice was based on their simplicity and their efficacy as a sentiment-laden source of data. To further clarify, we have included a reference to a cosine similarity analysis with a toy adverb dataset in the appendix, which supports our choice.
>
> > In Section 4.1, the authors find that the model employs a simple and interpretable algorithm to perform the task. However, this algorithm has not been supported by corresponding experiments…
>
> We understand your reservations regarding the claim in Section 4.1. We provided our experimental evidence for this interpretable algorithm in Appendix A.3, but did not make this prominent in the main body and it may have been missed. We have improved the main text with additional clarifications and a pointer to the relevant experiments in the appendix.
>
> Your feedback has been very helpful, and we hope our revisions adequately address your concerns!

---

> > ### Author Response · Authors · 2023-11-21
> >
> > Dear reviewer 6Ynr, since the end of the discussion period is on November 22 (in two days) we would like to gently remind you about our rebuttal to your comments. We hope that you can let us know if we have addressed your concerns! (Note that we will keep adding results for additional models throughout the remainder of the period). If there are any remaining questions or concerns, we would be happy to continue the discussion.

---

> > > ### Comment · Reviewer_6Ynr · 2023-11-22
> > >
> > > I am very grateful for the detailed response provided by the author, which has resolved my concerns. I will keep the original score of 6.

---

### Official Review · Reviewer_8vUR · 2023-10-31

**Soundness:** 2 fair
**Presentation:** 1 poor
**Contribution:** 2 fair
**Rating:** 3
**Confidence:** 3

**Summary:**

This paper proposes that pre-trained models have already learnt the notion of sentiment up to linear transformations in their intermediate layers. THe linear representation hypothesis suggests that large language models learn representations of text that are linearly related to meaning or related notions. In this work, the authors evaluate this idea for certain large language models (GPT-2 and Pythia) on several datasets, including toy datasets, SST and OpenWebTet, for the notion of sentiment.

To find the directions among intermediate layers of the language model, the authors propose to use simple techniques such as mean of the activation vectors, linear probing, PCA and DAS. They discover such directions for the toy datasets and evaluate it via correlation analysis. Similar analysis on the other datasets, including activation patching leads them to discover that sentiment is being aggregated on various punctuation tokens in the text, which they call summarization motif.

While containing potentially interesting ideas, I find the writing poor and hard to understand. A lot of terms are left undefined, the diagrams are essentially the only quantitative part of the work and the reader is left to decipher or understand a lot of the references by themselves. Please see weaknesses below.

**Strengths:**

- It's widely believed that large models learn activations that linearly capture conceptual latent variables. This work proposes that the sentiment notion is also captured linearly among the activations of intermediate layers.

- The idea of summarization motif that information aggregates at certain tokens, is also interesting.

**Weaknesses:**

- The writing seems extremely sloppy. For an experimental paper, many of the plots/discussions are not clear, especially for a broad ICLR audience. For example, what are the x- and y-axes of figure 2b?

- What are "sentiment activations" in GPT2-small, are these related to sentiment directions that are learnt via the 5 methods?

- I feel like the authors are using the term "causal" loosely, do they mean that the sentiment is sensitive to activation addition, and hence they deem it causal? Usually the term causal representations are reserved for representations that generate the data, not the other way around.

- As the authors note, attention heads are used for the analysis and a more thorough analysis should include activations at MLP layers too.

**Questions:**

Some questions were raised above.

- Figure 2 should probably be moved towards where it is discussed, instead of at the introduction.

- Consistency: Both "k-means" and "K-means" are used.

- In reproducibility statement, the words "Appendix Section" are both used for A.3.

---

> ### Author Response · Authors · 2023-11-17
>
> Thank you for your feedback on our paper. We appreciate the opportunity to clarify aspects of our work and refine its presentation.
>
> > The writing seems extremely sloppy. For an experimental paper, many of the plots/discussions are not clear, especially for a broad ICLR audience. For example, what are the x- and y-axes of figure 2b?
>
> Regarding the writing clarity, we regret any confusion and have extensively improved our paper’s clarity and readability in the latest revision. We have conducted a thorough review and made many small improvements, shown in red font for your convenience. We have brought key definitions forwards from the glossary into the main body and improved figure explanations, particularly for Figure 2b. We would also like to draw your attention to the helpful glossary in Appendix A.7.
>
> > What are "sentiment activations" in GPT2-small, are these related to sentiment directions that are learnt via the 5 methods?
>
> These are indeed highly related. We would like to draw your attention to the definition given in the opening text in Section 3:
>
> > We show that the methods discussed above (2.2) all arrive at a similar sentiment direction. We can visualise the feature being represented by this direction by projecting the residual stream at a given token/layer onto it, using some text from the training distribution. We will call this the ‘sentiment activation’.
>
> > I feel like the authors are using the term "causal" loosely, do they mean that the sentiment is sensitive to activation addition, and hence they deem it causal? Usually the term causal representations are reserved for representations that generate the data, not the other way around.
>
> By “causally significant,” we do not mean that the sentiment direction is sensitive to activation addition, but rather that causal mediation (i.e. activation patching) experiments demonstrate the causal effect that these directions exert on the models’ outputs in real-world contexts. We would like to draw your attention to methods section 2.3 Causal Interventions, where we outline exactly two causal methods: “Activation Patching” and “Ablations”. Then in results section 3.3 Causal Evaluation, the first sentence reads “We evaluate the sentiment direction using directional patching”. In the figure captions for our causal results, we preface 2d with “Directional patching results” and 2e with “Patching results”. From the opening to Section 4: “Through an iterative process of path patching (see Section 2.3)”. In the reproducibility statement: “Our causal analysis techniques of activation patching, ablation, and directional patching are presented in Section 2.3”.
>
> The activation addition results were not central to our claims and thus were relegated to the Appendix in Figure A.3. We regret that these may have seemed like our basis for using the term “causal”. To avoid any further confusion, in our latest revision we have removed the single reference to activation addition from the main body of the paper. We appreciate your attention to this detail and are open to any further suggestions or questions you might have.
>
> > As the authors note, attention heads are used for the analysis and a more thorough analysis should include activations at MLP layers too.
>
> Thank you for raising this point. We note that our core results focus on studying directions in the model’s residual stream (i.e. in the skip connections around layers), which are the sum of the outputs of *all* prior layers, including both attention and MLP layers, and thus includes the contributions of the MLP layers..
>
> We also provide more detailed circuit analysis that we agree would benefit from a broader examination that includes MLP layers. However, this is not central to our core contribution in this paper, and as such, though we have provided initial findings, we acknowledge this as a limitation and an avenue for future work. In addition, in our path patching experiments we observed that the MLPs were of relatively low importance to the models’ performance on the tasks we analyzed, a finding that parallels that of past circuit analyses such as that in Wang et al. ([https://arxiv.org/abs/2211.00593](https://arxiv.org/abs/2211.00593)) Since attention heads had a more significant effect on model performance, we focused on them in our circuit-level analyses.
>
> Your feedback is welcome, and we will make all necessary revisions to meet the high standards of the conference. We believe that the core contributions of our work hold significant value and, with your feedback, we aim to present them more clearly. Please let us know if we have addressed your objections, or if there are further changes you would like to see made to the paper.

---

> > ### Author Response · Authors · 2023-11-21
> >
> > Dear reviewer 8vUR,
> > Since the end of the discussion period is on November 22 (in two days) we would like to gently remind you about our rebuttal to your comments. We hope that you can let us know if we have addressed your concerns! If there are any remaining questions or concerns, we would be happy to continue the discussion.

---

> > ### Comment · Reviewer_8vUR · 2023-11-21
> >
> > I thank the authors for their thoughtful response. They have attempted to incorporate my concerns as well as the ones raised by the other reviewers. However, this has led to a significant rewriting of the paper already (all the red changes), the authors are working on adding more experiments and this revised work will likely need another careful evaluation.
> >
> > Therefore, I no longer feel comfortable recommending the current version for acceptance. I appreciate the authors trying to explain their ideas in the rebuttal but I still reserve my opinion that it seems like the work is borrowing ideas from lots of other works, making lots of (confusing) connections and is not particularly easy to read. The paper seems like it's written specifically for experts in the field and would require significant revisions for a large and diverse ICLR audience.

---

### Official Review · Reviewer_dUnj · 2023-11-01

**Soundness:** 2 fair
**Presentation:** 2 fair
**Contribution:** 4 excellent
**Rating:** 6
**Confidence:** 3

**Summary:**

Large language models (LLM) can perform zero-shot classification, such as sentiment analysis with a simple discrete prompt. This paper studied how large language models represented their sentiments. The authors argued that the sentiment is represented linearly, i.e., there is a vector pointing in the direction of positive (or negative) sentiment. In addition, the authors proposed the summarization motif -- a phenomenon in an LLM that aggregates information (about sentiment) at indiscriminative tokens (e.g., commas).

To prove the linear sentiment representation, the authors applied several approaches to extract the sentiment direction from a small toy dataset and performed directional activation patching. In the experiment on a subset of SST, the results showed that the direction was significant to the logits -- enough to flip the prediction up to 53.5% of the time.

To prove the summarization motif, the authors applied a similar patching method to the commas (or periods?) and compared it with patching all tokens. The results showed that only patching commas contributed to more than half of the drop in accuracy (38% to 18%) on a subset of SST. Additional analysis of another dataset showed that the further the relevant phrase in a prompt, the more apparent the summarization motif was.

**Strengths:**

The paper presented original findings on LLM behavior and sentiments. The paper applied existing tools to prob the LLM using a new format of experiments. The linear representation of sentiment in GPT was an interesting finding and added insights to the disentangled representations of unsupervised models. The summarization motif of the sentiment was also original. While we know that some heads LLMs (or smaller ones) tend to attend delimiters, we still do not understand why (Clark et al., 2019). The experiment result shown in Table 1b revealed an insight into this behavior.

Clark, K., Khandelwal, U., Levy, O., & Manning, C. D. (2019). What does BERT look at? An analysis of BERT's attention. arXiv preprint arXiv:1906.04341.

**Weaknesses:**

Although the paper presented original findings, I found a few issues in the experiment to support the claims and some inconsistent results.

1. Models. The authors claimed that the results were consistent across a range of models. However, I found mostly GPT2-small or Pythia in the experiment results -- one model for each, not repeated. While this is already an interesting finding, it has weak support for the claim. The author should either clarify this or lower the claim.

2. Linearity. The authors claimed that the sentiment was linearly represented. Still, the evidence was rather weak on the real dataset (SST), especially the vectors discovered by simple linear methods like PCA or logistic regression. The effect was more prominent than the random directions, but it did not support the linearity claim. While the authors explained that the direction was discovered from a small dataset, we could also explain that there was more than one direction or simply nonlinear sentiment space.

3. Distance. While the author experimented with a toy dataset and injected irrelevant content, I found the relationship between the "summarization" and the context length interesting. However, this could be made stronger -- the authors could find sentences in SST with varied lengths and commas to support the relationship.

4. Inconsistency. In the abstract, the authors wrote 76% and 36%, but these were not consistent with the text in Section 4.3.

**Questions:**

1. Can you clarify why the results were consistent across different models?

2. Can you explain why the logit flip in the abstract is doubled from the text?

---

> ### Author Response · Authors · 2023-11-17
>
> Thank you very much for the reference to Clark et al. which we have added to the introduction in our latest revision. We are also very grateful for your thoughtful comments in general!
>
> > Models. The authors claimed that the results were consistent across a range of models… author should either clarify this or lower the claim.
>
> Where possible, we have tried to run the same experiments across different models. In particular, the core experiments of patching the sentiment direction in different layers when evaluating on our real-world dataset (SST) were performed for 4 different sizes of GPT-2 and 4 different sizes of Pythia. We included a representative selection of these in Figure 2e. Some tasks could not be performed by smaller models (such as the mood stories) and other experiments were not performed on larger models due to compute time constraints (such as evaluating Pythia-2.8b on SST with patching at each layer).
>
> To add support to our claim of universality, we have repeated the experiments in Figures 2a and 2b for Pythia-1.4b, in order to provide a comparison with GPT2-small. These show that in both models, the different methods converge on a single direction which can classify sentiment on OpenWebText.  In addition, to provide additional support for our “summarization motif” results, we have added activation patching experiments for Pythia 6.9b to Appendix A.4.2, and are currently working on adding SST direction ablation experiments for other models to a future revision.
>
> > Linearity. The authors claimed that the sentiment was linearly represented… we could also explain that there was more than one direction or simply nonlinear sentiment space.
>
> Thank you for pointing this out; we agree that this claim could benefit from further clarification and we have added a section to the Appendix (A.2) in the revised version, where we include some text expressing similar limitations to the above for our linearity claim.
>
> Our core contribution lies in identifying a key sentiment direction that is causally influential and consistent across various tasks, as evidenced by our directional patching experiment's ability to leverage a sentiment direction identified from a toy dataset to effectively classify sentiment in more complex datasets like OpenWebText and Stanford Sentiment Treebank. While we acknowledge the potential for additional sentiment representations tailored to specific tasks or non-linear dimensions, our findings suggest the dominance of a primary, shared sentiment direction.
>
> We propose that this primary direction acts as a "first principal component of sentiment," and that there could be further more nuanced, task-specific directions. This concept echoes the "feature splitting" phenomenon described by Bricken et al. in “Towards Monosemanticity.” Although exploring these finer-grained directions is outside our current scope, we believe pinpointing this overarching sentiment direction is nevertheless a meaningful contribution to the literature.
>
> We would also like to draw your attention to Appendix Figure A.6, where we show that allowing DAS to learn a higher dimensional subspace does not improve the loss out of distribution. This supports the linear perspective, showing that it is hard to beat even by increasing the dimension of the DAS subspace when seeking a representation of sentiment across a sufficiently wide range of tasks.
>
> > Distance. While the author experimented with a toy dataset and injected irrelevant content, I found the relationship between the "summarization" and the context length interesting. However, this could be made stronger…
>
> Thank you for your interest in our result showing increasing summarization with distance. We would also love to see further study of this result, but we have struggled with a number of confounders in real-world datasets. As this is a very interesting phenomenon but not a central one to our core claims in this paper, we decided to merely flag the finding here and leave these complexities for future researchers.
>
> > Inconsistency. In the abstract, the authors wrote 76% and 36%, but these were not consistent with the text in Section 4.3.
>
> Thank you for raising this. In hindsight, we agree that reporting the same result with two different denominators was confusing. We reported the same result in terms of the 76% reduction in accuracy above “chance” (i.e. 50% for a balanced binary dataset) and in the body we reported this in terms of a drop of the 38% drop in raw accuracy (i.e. above 0%), resulting in a factor of two discrepancy. The same was true for the 36% and 18% accuracy drops reported with comma-only ablation. We have since revised our numbers reported in the abstract to only use the more conservative framing
>
> We are grateful for the opportunity to refine our work based on your valuable feedback and look forward to further discussion of these results.

---

> > ### Author Response · Authors · 2023-11-21
> >
> > Dear reviewer dUnj,
> > Since the end of the discussion period is on November 22 (in two days) we would like to gently remind you about our rebuttal to your comments. We hope that you can let us know if we have addressed your concerns! (Note that we will keep adding results for additional models throughout the remainder of the period). If there's any remaining questions or concerns, we would be happy to continue the discussion.

---

> > > ### Comment · Reviewer_dUnj · 2023-11-23
> > > **Thank you for your response**
> > >
> > > Thank you for your detailed response. I found that most of my concerns had been discussed and addressed to some extent. The additional experiments resolved the generality concern. As for the linearity concern, which is the central claim of the paper, I found the "first principal component of sentiment" was a bit weak compared to the claim, but the result in Figure A.6 was insightful.
> > >
> > > Based on this response and other reviews, I decided to increase my score to 6.

---

### Author Response · Authors · 2023-11-17

We’d like to thank the three reviewers for their thoughtful reviews. We are very excited to see that the reviewers found our two core results, the linearity and summarization of sentiment, to be deeply interesting. That is, we have demonstrated that sentiment is represented linearly in the residual stream across a range of sizes of GPT2 and Pythia models, and that these representations are not just stored on valenced words, but also on punctuation tokens with no inherent sentiment, summarizing the sentiment of the preceding clauses. We were grateful for feedback on how to clarify our findings and have made revisions to the paper based on this feedback. In particular, the main criticism which was rightly brought to our attention was the need for greater evidence of our claim of universality across models and datasets. We have addressed this with novel experiments, culminating in revisions listed under headings 1-2. We believe that these new experiments significantly improve the strength of our claim for a universal linear sentiment representation and widespread use of the summarization motif across a broad range of LLMs.

Here is a list of revisions:

1. Increasing range of models
    a. We repeated our comma summarization experiments for the ToyMoodStory dataset on the larger model Pythia-6.9b and found even stronger results! We have put these results in Appendix A.4.2.
    b. We have focused on Pythia results in Figure 2 and highlighted the similarity of these findings when studying a very different model, GPT2-small, in a new Appendix Figure A.6 (Section A.1.5).

2. Increasing range of data
    a. We have replicated the Stanford Sentiment Treebank directional ablation experiment on the IMDB reviews dataset. These prompts are significantly longer and thus the results better demonstrate that the direction captures sentiment at different levels, from single sentences to the level of entire documents. We have added these to Section 3.3.

3. Adding clarity to the writing
    a. We have expanded on the limitations of our linearity claim in Appendix A.2.
    b. We have defined path patching in method section 2.3.
    c. We have added more in-line definitions in the main text.
    d. We have added references to the relevant methods section where applicable.
    e. We expanded the Glossary (including ablation, circuit, path patching, sentiment activation, etc.).

4. Moving non-essential experiments to the Appendix
    a. We moved the negation experiment to Appendix A.1.4
    b. We have moved the reference to “activation addition” to Appendix A.1.2.

5. Added a reference to Clark et al in the introduction

In addition, we are currently working on experiments to extend our Stanford Sentiment Treebank summarization results to other model families and hope to have the results ready before the end of the rebuttal period.

---

> ### Author Response · Authors · 2023-11-23
>
> The consensus amongst the reviewers has been that the two core results of our paper (that is, 1. That we have found a causal, linear direction representing sentiment, and 2. That models aggregate clause-level sentiment information at punctuation, forming a partial information bottleneck) are very interesting, but that we needed to provide further support of their universality. Reviewer 8vUR also found some of our writing unclear. In a timely manner, we have provided novel experimental results in both new models (Pythia-6.9b and some “summarization motif” evidence in Mistral-7b) and a new dataset (IMDB) to more firmly establish our findings. We have also expanded the glossary and in-line definitions to ensure that our work is comprehensible to a diverse ICLR audience.
>
> Based on the feedback of reviewer dUnj, we have provided further evidence of universality and limitations to our linearity claim, improving the rigor and robustness of our paper. We thank them for noting that we were able to address most of their concerns, and for their increase in their indicated rating.
>
> We would also like to thank reviewer 6Ynr for acknowledging our detailed response and that their concerns were resolved.
>
> After reviewer 8vUR provided some criticism of our writing and clarity, we made changes that we believe significantly improved the polish on our paper. Afterwards, the reviewer argued that our changes were too significant to be re-reviewed and justify a re-submission, but we respectfully disagree and invite the area chair to review the revised manuscript and decide if the changes (conveniently shown in red text) are unreasonably large.

---

### Meta-Review · Area_Chair_e8Lo · 2023-12-07

**Metareview:**

This paper investigates the degree to which LMs store sentiment information in a way that is linearly decodable. To this end, the authors include in their evaluation two LMs (GPT-2 and Pythia [1.4b and 2.8b; 6.9b added in response period]), and use existing methods to attempt to recover a "sentiment direction". Specifically, this includes both 'probing' methods and causal interventions. The results indicate that there is a "sentiment" direction in such models (extracted over templated text, but validated on SST-2), and moreover that sentiment appears to be "summarized" on, e.g., punctuation tokens.

There are some cool findings in this paper! But I think its marred by lackluster presentation (an issue raised by 8vUR and 6Ynr as well). The authors addressed some of this in revision, but having now read the paper myself, I think it remains difficult to follow many key points and findings. The plots (which serve to report the main findings) are difficult to read, for example. In my view, the paper would benefit from focussing on a subset of the interpretability methods used here (e.g., LR) together with the causal analysis, this would simplify presentation and interpretation of the findings, and additional results could be reported as supplementary material.

Another issue is that it is difficult to say precisely what was done, technically, for some of the analyses. For instance, I think 2.3 is sufficiently brief that it if one is not reasonably familiar with these methods, they are unlikely to be able to follow the main findings, and this may hamper the impact of the work when presented to a slightly more general audience like ICLR. The circuit analysis (4.1) is interesting, but practically no details are offered how this was carried out.

**Justification For Why Not Higher Score:**

While there are some cool results here, this work feels undercooked and suffers from suboptimal presentation which makes it unnecessarily difficult to follow.

**Justification For Why Not Lower Score:**

N/A

---

### Decision · Program_Chairs · 2024-01-16

Reject